# CRISPR-Cas-based identification of a sialylated human milk oligosaccharides utilization cluster in the infant gut commensal *Bacteroides dorei*

Sivan Kijner [1], Dena Ennis [1], Shimrit Shmorak[1], Anat Florentin[1,2] & Moran Yassour[1,3] ✉

The infant gut microbiome is impacted by early-life feeding, as human milk oligosaccharides (HMOs) found in breastmilk cannot be digested by infants and serve as nutrients for their gut bacteria. While the vast majority of HMO-utilization research has focused on *Bifidobacterium* species, recent studies have suggested additional HMO-utilizers, mostly *Bacteroides*, yet their utilization mechanism is poorly characterized. Here, we investigate *Bacteroides dorei* isolates from breastfed-infants and identify that polysaccharide utilization locus (PUL) 33 enables *B. dorei* to utilize sialylated HMOs. We perform transcriptional profiling and identity upregulated genes when growing on sialylated HMOs. Using CRISPR-Cas12 to knock-out four PUL33 genes, combined with complementation assays, we identify GH33 as the critical gene in PUL33 for sialylated HMO-utilization. This demonstration of an HMO-utilization system by *Bacteroides* species isolated from infants opens the way to further characterization of additional such systems, to better understand HMO-utilization in the infant gut.

Following birth, the dynamic and complex process of the infant gut microbial colonization is initiated[1–4]. The main factors that determine the composition of the infant gut microbiome are delivery mode[2,5–7] and infant feeding (formula vs. breast-milk)[5,7–9]. Specifically, the third-largest solid component in breast-milk is human milk oligosaccharides (HMOs), which are a family of glycans indigestible by infants[10,11], left to be consumed by bacteria in the infant's gut. HMOs have a variety of health benefits, such as being decoy receptors for pathogens, as they resemble receptors found on the infant's intestinal epithelium, with which pathogens can interact[12–16]. HMOs can be divided into three main groups based on their structure: fucosylated, sialylated, or neutral sugars. Sialylated sugars have special importance in infant health and

are able to shift metabolism to increase lean body mass and improve bone morphology[17,18]. Also, offspring of mice producing milk deficient in sialylated HMOs were found to have defects in nervous system development and neuronal patterning, particularly in areas associated with executive functions and memory[19].

One of the most studied roles of HMOs in infancy is the promotion of a healthy infant gut microbial community, advantageous for future health[20–23]. HMOs support the colonization of beneficial gut bacteria that can break them down[24] and most research in the past two decades has focused on *Bifidobacterium* species[25–41], which are considered as the hallmark HMO-utilizers in the breastfed-infant gut microbiome. However, recent evidence suggests that *Bifidobacterium* species are

[1]Microbiology & Molecular Genetics Department, Faculty of Medicine, The Hebrew University of Jerusalem, Jerusalem, Israel. [2]The Kuvin Center for the Study of Infectious and Tropical Diseases, Institute for Medical Research Israel–Canada, Faculty of Medicine, Hebrew University of Jerusalem, Jerusalem, Israel. [3]The Rachel and Selim Benin School of Computer Science and Engineering, The Hebrew University of Jerusalem, Jerusalem, Israel. ✉ e-mail: Moranya@mail.huji.ac.il

not the only commensal HMO-utilizers in the infant gut. Studies investigating the infant gut microbiome composition report frequent cases (up to 90%) in which breastfed-infants do not harbor any *Bifidobacterium* species throughout their breastfeeding period[2,42,43]. There are also significant correlations between specific HMO levels and the relative abundance of other genera, such as *Bacteroides*, *Parabacteroides*, and *Lactobacillus*[28,44–46]. In addition, computational annotation of genomes of common infant gut microbes points towards some genes as potential HMO-utilizers[2,43,47], mostly from the *Bacteroides* genus. Taken together, all these suggest that *Bacteroides* species are also candidates for HMO utilization.

*Bacteroides* are known degraders of complex versatile glycans[48–51], using thousands of enzyme combinations[52], namely glycoside hydrolases (GHs)[53]. Their glycan utilization repertoire is determined by the variety of polysaccharide utilization loci (PULs) present in their genome[54,55]. A PUL is a gene cluster containing jointly-regulated genes that encode the machinery necessary for the breakdown of a specific sugar. *Bacteroides* species have developed multiple strategies for utilizing HMOs with varying levels of efficiency, as was demonstrated for type strains of *B. fragilis*, *B. thetaiotaomicron* and *B. vulgatus* on both a pool of HMOs[56] and individual fucosylated HMOs[57]. Some of the mechanisms have been further identified, for example, that *B. thetaiotaomicron* and *B. fragilis* utilize both host mucin-O-glycans and HMOs using a similar set of upregulated genes[56]. For fucosylated HMOs, fucosidases from *Bacteroides* were profiled and showed versatility in efficiency and substrate bond specificity[58]. Finally, for sialylated sugars, one of *B. thetaiotaomicron*'s GH33 sialidases was crystallized and characterized[59], unraveling a wide binding groove accounting for a broad substrate specificity, including various sialylated HMOs[60].

While these pioneering works lay the foundation for understanding HMO utilization by *Bacteroides* species, further research is needed to extend the mechanistic characterization to additional infant gut commensal strains and specific types of HMOs. Commonly, mechanistic analyses rely on genetic manipulation of tested strains, mainly used in type strains that have well-characterized genomes, functions, and systems, rather than on natural isolates that differ (to an unknown extent) from their closest type strains. *Bacteroides* research thus far has focused on the *B. thetaiotaomicron* type strain[61], which requires a mutant parental strain[62–65] for allelic-exchange-based genetic manipulation[66–68]. However, this commonly used method suffers from several limitations, mainly low efficiency (especially in isolates). Focusing on natural isolates, generating a parental strain for each isolate is unlikely, thus additional approaches should be implemented. Finally, the genetic tools developed in *E. coli* tend to be incompatible with the transcription-translation machinery of *Bacteroides*, making the development of novel genetic manipulation systems even more challenging. Luckily, recent adaptations of a CRISPR-based system for gene knockout in *Bacteroides*[69,70] paved the way for specific, markerless, genomic editing of *Bacteroides* genomes, which is efficient not only for type strains but also for natural isolates.

Here, we implemented the recently developed CRISPR-Cas system to genetically manipulate natural isolates of *Bacteroides* from the infant gut. In *B. dorei*, we identified PUL33 as a necessary gene cluster for sialylated HMO utilization and specifically the GH33 as the critical gene for this purpose within this cluster. Finally, expressing *B. dorei*'s PUL33 genes in bacteria that cannot utilize sialylated HMOs, enabled them to grow on these sugars, demonstrating the sufficiency of this gene cluster.

## Results

### *B. dorei* and *B. vulgatus* grow on HMOs with varying efficiency

In our search for non-*Bifidobacterium* HMO-utilizers, we started with two *B. dorei* strains that we recently isolated, which were shown to grow on various HMOs[71]. Before we continued to mechanistically

examine the HMO utilization process in these strains, we first explored the generality of this phenotype in additional natural isolates. Towards this end, we isolated two additional *B. dorei* strains and six *B. vulgatus* strains (which are closest taxonomically in the *Bacteroides* species tree[72]) from breastfed-infant stool samples. We grew the isolates on minimal media (MM) supplemented with a single HMO representing all three structural groups; Fucosylated HMOs: 2-Fucosyllactose (2'-FL) & Difucosyllactose (DFL); Sialylated HMOs: 3-Sialyllactose (3'-SL) & 6-Sialyllactose (6'-SL); and neutral HMOs: Lacto-N-Tetraose (LNT) & Lacto-N-neotetraose (LNnT). We examined the growth kinetics of all 10 isolates on these sugars compared to glucose and lactose, which served as a control.

Vast majority of isolates were able to utilize the tested HMOs with varying degrees of efficiency (Supplementary Fig. 1). For all isolates, growth on fucosylated HMOs was inferior compared to growth on sialylated or neutral HMOs, both in terms of the lag phase length and the peak-OD value. This finding was surprising as fucosidases from GH families 29 and 95 (that participate in fucosylated-HMO breakdown[73]) are more prevalent among *B. dorei* and *B. vulgatus* compared to specialized sialidases from the GH33 family (that participate in sialylated-HMO breakdown[73]). When comparing the average counts of enzymes per genome, we found there are 8.82 average GH29 fucosidases and 4.14 GH29 fucosidases per genome, compared to only 2.14 average GH33 per a *B. dorei* or *B. vulgatus* genome[74] (see the "Methods" section).

Hereafter, we chose to focus on *B. dorei* isolates growing on sialylated HMOs due to a number of reasons. First, sialylated HMOs contribute to infant health and their effect is mediated by the gut microbiome[17–19] (see the "Introduction" section). Second, the consistent growth dynamics on the two tested sialylated HMOs (3'-SL, 6'-SL), together with the unique terminal residue of sialic acid suggest a specialized pathway or mechanism to be discovered, which is specific for sialylated sugar structures. Lastly, *B. dorei*'s growth on sialylated HMOs is overall comparable to glucose and lactose, especially in contrast to the poor growth on fucosylated structures, suggesting that sialylated HMOs serve as an effective carbon source for the species *B. dorei*.

### PUL33 is upregulated in *B. dorei* in response to sialylated HMOs

Differential expression of genes under various conditions can highlight the enzymes, mechanisms, and pathways enabling the utilization of specific carbon sources. Here, we re-analyzed previously published[71] RNA-sequencing data of one of our *B. dorei* isolates when growing on MM supplemented with the individual HMOs described earlier. The whole genome transcriptional profiling analysis revealed that a specific gene cluster called PUL33 is significantly upregulated when growing on sialylated HMOs compared to glucose (fold-change > 2, padj < 0.001; Fig. 1A, B). Of the remaining 65 PULs in *B. dorei*'s genome[74], none were completely upregulated as PUL33 was, rather only some of their genes were upregulated (Fig. 1A, Supplementary Fig. 2). The upregulation of PUL33 was specific to sialylated HMOs and was not evident on fucosylated or neutral HMOs (Fig. 1B, Supplementary Fig. 3), suggesting that PUL33 has a distinct role in the utilization of the sialylated HMOs tested (3'-SL, 6'-SL).

To further pin-point the specific genes that play a role in the breakdown of sialylated HMOs, we examined the individual genes of PUL33. One of the first steps in utilizing 3'-SL and 6'-SL is releasing the sialic acid residue, making the lactose core available for further breakdown. This function is performed by sialidases belonging to the glycoside hydrolase (GH) family 33[53]. Indeed, the gene encoding a GH33 sialidase was among the most upregulated genes in the entire genome and specifically the highest in PUL33, with a 50.21 and 58.49-fold increase in expression when growing on 3'-SL and 6'-SL, respectively (Fig. 1A, B). In the current annotation of *B. dorei*, this is the only GH33 gene. However, previous versions of *B. dorei*'s

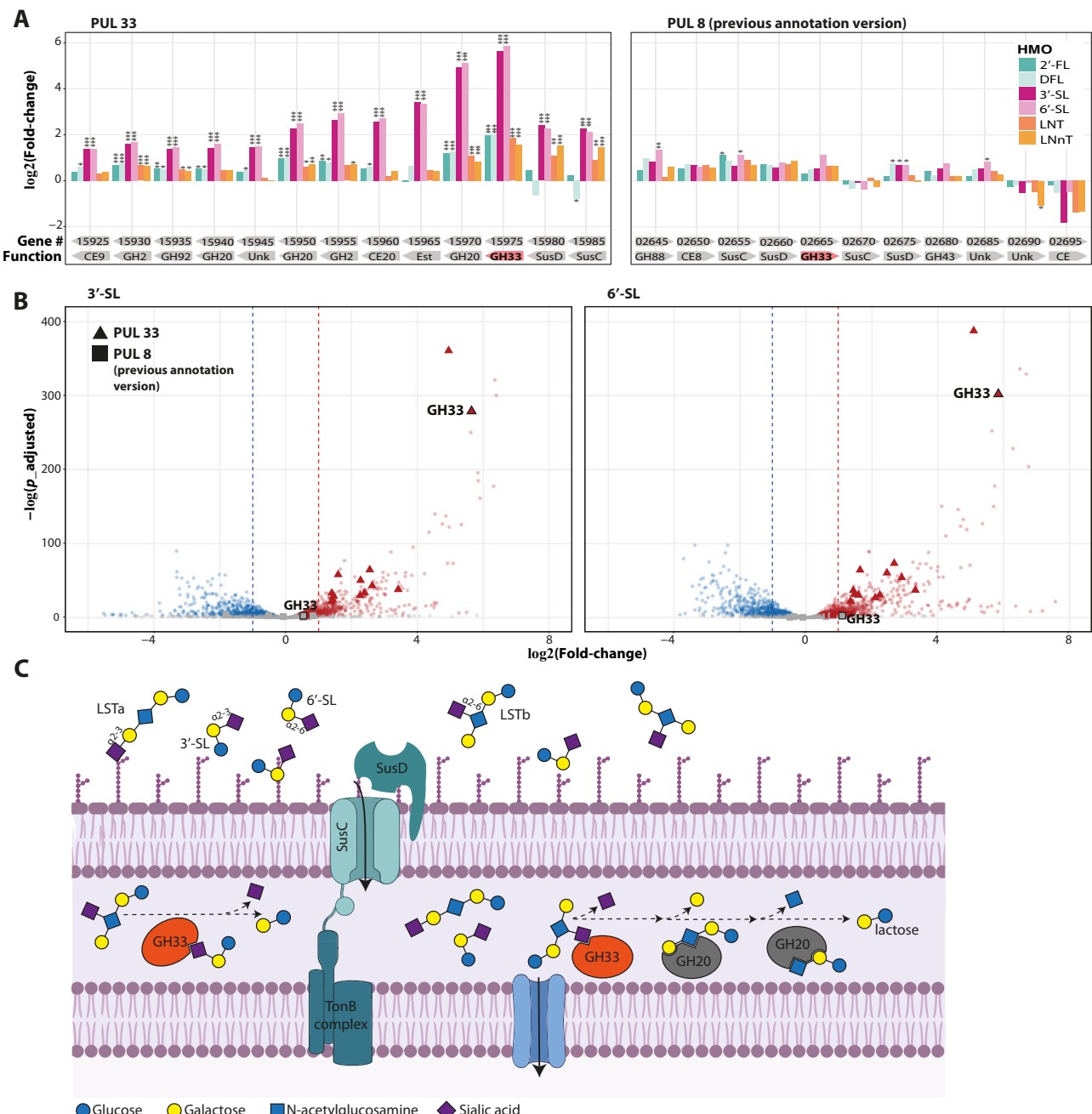

**Fig. 1 | *Bacteroides dorei* upregulates PUL33 in response to sialylated HMOs.**
Differential gene expression data comparing minimal media supplemented with various HMOs (2'-FL, DFL, 3'-SL, 6'-SL, LNT, LNnT) to glucose. **A** Log2 fold-change of genes encoded in polysaccharide utilization locus (PUL) 33 and the previous PULDB annotation of PUL8 in *B. dorei*. The gene GKD17_02665 was not annotated as GH33 in the subsequent version. Asterisks indicate significance as outputted by DESeq2 using the Wald test, after correcting for multiple hypothesis using the Benjamini and Hochberg method ($p$_adjusted *<0.05; **<0.01; ***<0.001). For all genes in PUL33 $p$_adjusted values were lower than $8.26*10^{-10}$ when comparing expression on sialylated HMOs compared to glucose. **B** Volcano plots of gene expression on sialylated HMOs, showing the fold-change (log2, $x$-axis) vs. the differential significance (-log2($p$_adjusted), $y$-axis), with highlighted genes from PUL33 and the previous annotation of PUL8. The statistical analysis is the same as in (**A**). **C** Schematic of sialylated HMO utilization by the core genes in *B. dorei*'s PUL33. GH: Glycoside hydrolase. Figure cartoon was created with BioRender.com [https://www.biorender.com/]. Source data are provided as a Source Data file.

assembly in the Polysaccharide-Utilization Loci DataBase (PULDB) included an additional GH33 sialidase named GKD17_02665 found in a previous version of PUL8[74]. The expression of the previously-annotated GH33 was not upregulated at all in our data (Fig. 1A, B). The lack of upregulation of GH33-PUL8 under the tested growth conditions supports the notion that GH33-PUL33 in the main sialidase in the context of 3'-SL and 6'-SL, however, does not rule out its potential role in utilizing other sialylated HMOs not tested here.

## PUL33 is necessary for sialylated HMOs utilization in *B. dorei*

In order to test if PUL33 is necessary for the utilization of sialylated HMOs, we aimed to assess the growth of *B. dorei* on minimal media containing 3'-SL and 6'-SL, while lacking specific genes in PUL33. It is not straightforward to knock-out a genomic region from a natural isolate of *Bacteroides*[62–67] and for this purpose, we used a CRISPR-based system recently developed utilizing Cas12a (Fig. 2A)[69]. As the CRISPR-based system requires sensitivity to erythromycin, we implemented it on an erythromycin-sensitive *B. dorei* isolate from our strain collection

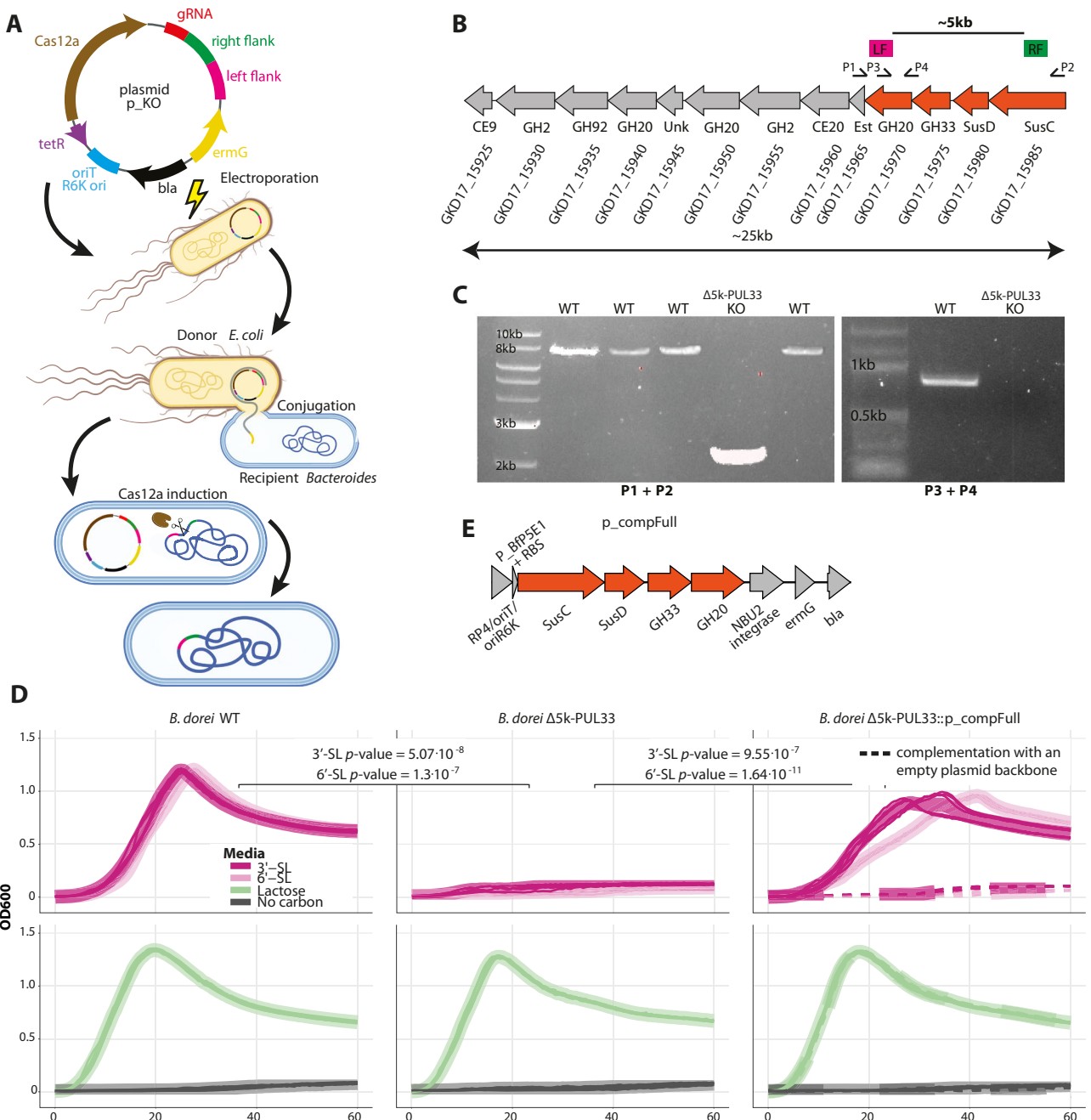

**Fig. 2 | PUL33 is necessary for sialylated HMO utilization in *B. dorei*. A** Overview of the CRISPR-cas12-based genetic manipulation system, including the transformation of *E. coli* and conjugation to the recipient *Bacteroides*. Figure cartoon was created with BioRender.com [https://www.biorender.com/]. **B** PUL33 gene structure with GenBank locus tags, highlighting the regions of homology (LF: left flank, pink; RF: right flank, green), with colors matching to the plasmid in (**A**) and primer locations (P1–P4). The genes that are nonfunctional in the mutant are shaded in orange. SusC SusC-like TonB-dependent transporter, SusD SusD-like cell-surface glycan-binding protein, GH glycoside hydrolase, CE carbohydrate esterase, Est Sialate O-acetylesterase, Unk unknown function. **C** PCR results from wild-type (WT) colonies, representing failed Cas12a inductions in the attempt to create a mutant and a single mutant (KO) colony, with shorter PCR product (lacking the removed genes). **D** Growth curve plots of *B. dorei*, *B. dorei* Δ5k-PUL33 and *B. dorei* Δ5k-PUL33::p_compFull. Growth curves were measured in minimal media supplemented with sialylated HMOs (3'-SL, 6'-SL; dark and light pink, respectively), lactose (positive control; green) or no carbon media (negative control; gray), 0.5% weight/volume. Thick lines represent the average of two biological replicates, each consisting of three technical replicates, which are represented by thin lines ($n = 6$). $p$-values were calculated using a paired two-sided $t$-test at 24 h. **E** Structure overview of the plasmid p_compFull. Source data are provided as a Source Data file.

and confirmed by RT-qPCR that it also upregulates PUL33 upon growth on sialylated HMOs (Supplementary Fig. 4A). Additionally, we performed whole genome sequencing of both isolates using long-read Oxford Nanopore Technology and generated a synteny map comparing them to the type strain *B. dorei* DSM 17855. The sequencing results revealed a high degree of similarity (average nucleotide identity range

98.92–99.04%) between the two isolates and the type strain, including the conservation of the entirety of PUL33 (Supplementary Fig. 4B).

Next, we constructed a *B. dorei* mutant, lacking 5000 bp from PUL33, which renders the genes encoding GH20, GH33, SusD, and SusC unfunctional (Fig. 2B). We chose to delete this specific region as it contains the transport system and the unique GH33 sialidase which

was the top upregulated gene in the transcriptional analysis (Fig. 1A). To validate the mutant, we performed two PCR reactions: First, using primers located before and after the deletion, showing that the PCR product is 5000 bp shorter in the mutant strain compared to the wild type (WT) strain (P1, P2 in Fig. 2C); Second, with one primer located within the deleted region, demonstrating a product at the predicted length for the WT and a lack of product for the mutant strain (P3, P4 in Fig. 2C). We also confirmed the *B. dorei* knock-out (KO) using sanger sequencing.

The KO mutant, designated *B. dorei* Δ5k-PUL33, was able to grow as well as the WT on the fucosylated and neutral HMOs (Supplementary Fig 5), however, it could not grow at all on any of the sialylated HMOs tested ($p$-values = $5.07 \times 10^{-8}$ and $1.3 \times 10^{-7}$ for 3'-SL and 6'-SL, respectively; Fig. 2D). To further validate our phenotype, we performed a complementation assay of the four deleted genes, using a previously described pNBU_erm integration vector[75], containing the desired genes under the constitutively expressed P_BfP5E1 promoter[67] (p_compFull; Fig. 2E). The complemented mutant, expressing the genes in a single copy from a heterologous genomic location, had restored the ability to grow on the sialylated HMOs (Fig. 2D, right). The complemented mutant reached a lower OD than the WT strain, possibly due to the expression of the complemented genes from the plasmid's *att1/2* integration site, rather than their endogenous genomic location. Taken together, these results indicated that the phenotype matches the genotype and further enhanced the essentiality of PUL33 in sialylated HMO utilization in *B. dorei*.

## The core genes in PUL33 enable sialylated HMO utilization in additional *Bacteroides* species

Next, we sought to examine whether the mere expression of GH20, GH33, SusD, and SusC from PUL33 is sufficient to drive sialylated HMO utilization in additional *Bacteroides* species. We selected two *Bacteroides* strains: *B. uniformis* CL03T12C37 and a *B. stercoris* isolate, that were unable to grow on sialylated HMO (Fig. 3A, Supp. Fig 6A) and do not encode any PUL33 homologs (see the "Methods" section). We then conjugated them with an *E. coli* strain containing the four core genes on p_compFull. While both wild-type strains did not utilize sialylated HMOs, the transconjugants, which have integrated p_compFull into their genome, were able to grow on these sugars. The *B. uniformis* strain was able to fully grow on the sialylated glycans with the addition of the PUL33 core genes, to an OD600 comparable to that of lactose (Fig. 3A) and the *B. stercoris* isolate exhibited significant, yet modest growth on the sialylated glycans (Supplementary Fig 6A). Although the growth of *B. stercoris*::p_compFull on sialylated HMOs is not robust, it is noteworthy, especially in the context of *B. stercoris*' growth on lactose, which is moderate as well (Supplementary Fig 6A). These results indicate that the four core genes (GH20, GH33, SusD, and SusC) are sufficient to cleave the sialic acid group from sialylated HMOs and thereby enable the growth of these sugars. Furthermore, this biochemical function is not entirely dependent on the specific cellular environment of *B. dorei*.

## GH33 in PUL33 is the key gene in the ability to utilize sialylated HMOs

Going back to *B. dorei*'s functionality and following the full complementation that restored the phenotype in *B. dorei* Δ5k-PUL33, we sought to explore whether all four complemented genes are necessary for PUL33's function by partial complementation assays. We constructed three partial complementation plasmids (p_compPartial1/2/3) each complementing a different set of genes from the four core ones (Fig. 3B) and conjugated them into the mutant *B. dorei* strain (hereby annotated as *B. dorei* Δ5k-PUL33::p_compPartial1/2/3). We confirmed the expression of the complemented genes using reverse-transcription quantitative PCR (RT-qPCR; Supplementary Fig 7). Of the three complemented strains, *B. dorei* Δ5k-PUL33::p_compPartial1 and *B. dorei*

Δ5k-PUL33::p_compPartial3 were able to grow well on sialylated HMOs, while *B. dorei* Δ5k-PUL33::p_compPartial2 was not able to grow at all on this media (Fig. 3C).

The plasmids that restored growth were only the ones that contained the GH33 gene (GKD17_15975), while the plasmid that did not restore growth was the one that lacked this gene (Fig. 3B, C). These results suggest that GH33 is critical for the utilization of sialylated HMOs in *B. dorei*, while the other genes of this cluster can be substituted by other enzymes encoded in *B. dorei*'s genome. We further validated this result by complementing *B. dorei* Δ5k-PUL33 with GH33 only (p_compPartial4), which enabled growth on sialylated HMOs (Fig. 3B, C). For *B. uniformis* CL03T12C37 as well, complementation with plasmids containing the GH33 gene (p_compPartial1/3/4) allowed modest growth on 3'-SL and 6'-SL, whereas complementation with the p_compPartial2 plasmid that lacks GH33 did not convey this ability (Fig. 3B, C). However, for the *B. stercoris* isolate, none of the partial complementation plasmids were sufficient to drive significant growth on sialylated HMOs (Supplementary Fig 6B, C), suggesting additional functions are missing in the *B. stercoris* cellular environment to enable even a modest growth.

## Predicted structure of *B. dorei*'s GH33 suggests a broad substrate specificity

To have a broader understanding of the phylogenetic context of the GH33 gene, we next performed a phylogenetic analysis of a subset of GH33 protein sequences deposited in the Carbohydrate-Active EnZymes (CAZy) database[76] (see the "Methods" section; Supplementary Fig 8). We found that most *Bacteroides* GH33 sequences cluster together, adjacent to other bacterial sialidases, with the GH33 proteins of *B. dorei* and *B. vulgatus* ATCC 8482 the most similar (Supplementary Fig 8), with a 97.44% identity (Supplementary Fig 9A). Additionally, some *Bacteroides* GH33 sequences were clustered closer to the eukaryotic branch as was previously described[77].

When zooming into the *Bacteroides* branch, there is a single *B. thetaiotaomicron* VPI 5482 GH33 protein for which the structure has been characterized using crystallography (GenBank accession AAO75562.1; PDB code 4BBW)[59]. We compared the predicted structure of our *B. dorei* isolate's GH33 (using AlphaFold;[78] pLDDT = 95.1; Fig. 4A) to the known structure of *B. thetaiotaomicron*'s GH33. Other than the first 23 N-terminal residues of *B. dorei*'s GH33, for which AlphaFold could not predict structure, the overall structure of the proteins is very similar (RMSD = 0.444 Å for a 521 out of 546 residues alignment; Fig. 4B, Supplementary Fig. 9A, see the "Methods" section). The six-bladed β-propeller catalytic site of GH33 is conserved between *B. thetaiotaomicron* and *B. dorei*, with 100% identity for the catalytic residues and additional residues that were found to interact with the substrate in the active site (Fig. 4C, Supplementary Fig 9B). The wide-open groove around the active site in *B. thetaiotaomicron* GH33 sialidase was found to be responsible for the broad substrate specificity of this enzyme[59]. Of the three residues that contribute to this wide groove, two were conserved in *B. dorei* while one differed between the hydrophobic Alanine and Valine (residue 228; Fig. 4C). This high conservation of the catalytic site, its neighboring residues, and the wide topology around the active site suggests that like *B. thetaiotaomicron* VPI 5482, *B. dorei*'s GH33 could also break down various sialylated glycans found in the infant gut.

## Discussion

In this study, we explore the growth kinetics of *B. dorei* and *B. vulgatus* isolates on various HMOs and implement a CRISPR-Cas system to genetically manipulate a natural isolate of *B. dorei* from a breastfed-infant stool sample. Using this system, we find that a single polysaccharide utilization locus called PUL33 is necessary for sialylated-HMO utilization in *B. dorei*. We demonstrate that the first four genes encoded in PUL33−GH20, GH33, SusD and SusC−provide the ability to

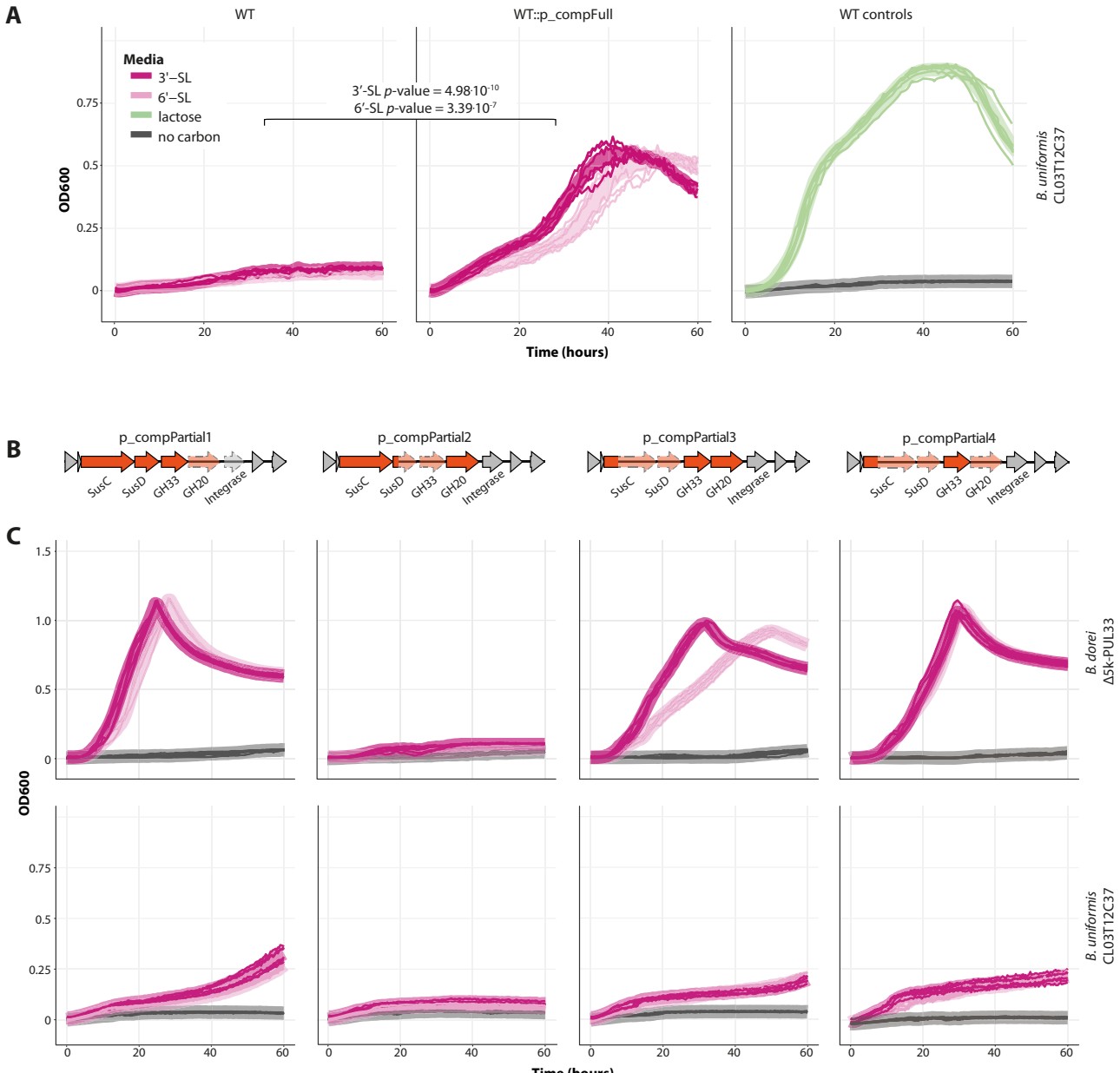

**Fig. 3 | Complementation assays to identify key genes for sialylated HMO utilization in PUL33. A** Growth curve plots of *B. uniformis* CL03T12C37 in minimal media supplemented with sialylated HMOs (3′-SL, 6′-SL; dark and light pink, respectively), lactose (green), or water (no carbon; gray), 0.5% weight/volume. Thick lines represent the average of two biological replicates, each consisting of three technical replicates, which are represented by thin lines (*n* = 6). *p*-values were calculated using a paired two-sided *t*-test at 50 h. This strain gained the ability to grow on sialylated HMOs once conjugated with p_compFull. **B** Structure overview of the plasmids p_compPartial1/2/3/4. **C** Growth curve plots of *B. dorei* Δ5k-PUL33 and *B. uniformis* CL03T12C37 complemented with p_compPartial1/2/3/4, in minimal media supplemented with sialylated HMOs (3′-SL, 6′-SL), 0.5% weight/volume. The colors and replicates are the same as in (**A**). Source data are provided as a Source Data file.

utilize sialylated HMOs in additional *Bacteroides* strains that are unable to grow on these sugars. Finally, utilizing partial complementation assays, we show that GH33 is the critical gene of PUL33 with respect to utilizing sialylated HMOs.

*Bacteroides* isolates exhibit robust growth on sialylated and neutral HMOs, whereas for most isolates the growth on fucosylated HMOs is partial, both in terms of a long lag phase and a low maximal OD600 value. Despite the fact that *Bacteroides* encode more fucosidase enzymes than sialidase enzymes in their genome, their ability to utilize fucosylated HMOs is poor. In *Bifidobacterium* species, well-known fucosylated-HMO utilizers[79], it was shown that the mere presence of fucosidase genes does not necessarily translate phenotypically to

better utilization, probably due to a strict substrate specificity[80,81]. Additional factors that can account for the inferior growth of *Bacteroides* on fucosylated HMOs are the lack of specific transporters for fucosylated HMOs, non-functional genes, or the inability to properly transcribe or translate genes in the pathway for the utilization of fucosylated HMOs.

From an evolutionary perspective, it seems like *Bacteroides* and *Bifidobacterium* species complement each other in their ability to utilize HMOs. Corresponding to the previous characterization of sialic acid consumption by *Bacteroides* species[82], they are adaptive to the utilization of sialic acid-containing glycans, such as the glycoproteins in the mucus layer and sialylated HMOs. *Bifidobacterium* species,

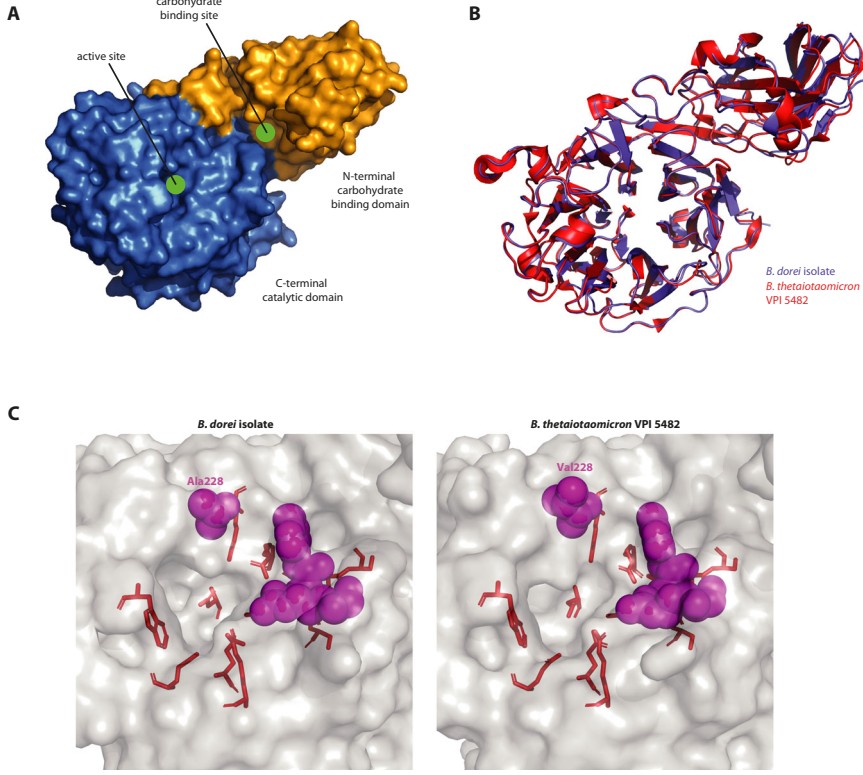

**Fig. 4 | Predicted structure of *B. dorei*'s GH33 resembles the crystallized GH33 sialidase of *B. thetaiotaomicron*. A** 3D structure of *B. dorei*'s GH33 sialidase, as predicted by AlphaFold based on the protein sequence, excluding the first 23 residues for which the pLDDT values are low. The active site and carbohydrate-binding site are indicated by a green circle. **B** Alignment of GH33 protein structures of *B. dorei* (predicted, blue) and *B. thetaiotaomicron* VPI 5482 (crystallized, red; PDB code 4BBW). The first 23 residues from *B. dorei*'s enzyme were excluded similarly to

(**A**). **C** The wide-open substrate-binding groove of the active site in *B. dorei* (left) and *B. thetaiotaomicron* VPI 5482 (right). Residues essential for activity are highlighted in red and residues responsible for laying out the wide-open groove are highlighted in magenta. Of the latter group of residues, although one of the three is not conserved between *B. dorei* and *B. thetaiotaomicron* (Ala228 and Val228, respectively), the wide topology of the groove is preserved.

---

however, are superior in their fucosylated-HMO utilization[79], perhaps at the expense of their sialylated-HMO utilization. This interesting dissimilarity in carbon preference could indicate that different species diversified to occupy various niches and utilize the variety of available carbon sources in the environment.

Focusing on our *B. dorei* isolate, we find that PUL33 is necessary for sialylated-HMO utilization. The conservation of PUL33 across all *B. dorei* and *B. vulgatus* genomes deposited in the PULDB database[74] suggests that PUL33 is important for the breakdown of sialylated HMOs. For *B. thetaiotaomicron* VPI 5482, a commonly used type strain, the closest PUL33 homolog is PUL9, experimentally shown to be involved in degrading sialic acid-containing mucins and HMO utilization[50,56]. While the GH33 protein sequence in the *B. dorei* and *B. thetaiotaomicron* clusters is similar (67.65%), their SusC transporters are only 20.67% alike (Supp. Fig 9C), suggesting the transporters have evolved to recognize varying substrates, or that these two clusters evolved independently to perform similar functions in different ways.

Interestingly, in the complementation experiments of *B. dorei*, we observe two growth pattern types: one resembling the WT strain growth (Fig. 3C; p_compPartial 1 and 4) and the other more delayed, with diverging growth patterns for 3'-SL and 6'-SL (Figs. 2D; 3C; p_compFull and p_compPartial 3). These differences are intriguing and could represent additional underlying principles that can be further characterized in follow-up studies. Additionally, we demonstrate that *B. dorei* can grow on 3'-SL and 6'-SL even in the absence of the *SusCD* transport system. This phenotype could potentially be explained by alternative transporters or by non-specific entrance of the small molecules of 3'-SL and 6'-SL into the cell through pores. These hypotheses are supported by a study on *B. thetaiotaomicron*, showing

delayed, yet successful, growth on ribose even in the absence of the ribose-specific transporters[55], suggesting that some simple sugars can indeed enter the cell non-specifically. Moreover, sialic acid is known to enter microbial cells by diffusion through nonspecific porins (OmpF, OmpC) in addition to the active TonB-dependent transport system found in *Bacteroides*[83,84], although this was not shown thus far for small molecules containing sialic acid.

Finally, when we zoom into GH33, the key gene in PUL33 responsible for the cleavage of the sialic acid residue from the HMO backbone, we find that *B. dorei*'s GH33 protein sequence is very similar to that of its close relative *B. vulgatus* and less so to *B. thetaiotaomicron* (97.44% vs. 67.65% similarity; Supplementary Fig. 9A). An interesting common feature of these GH33 proteins is a wide-open groove around the active site, which differs from other sialidases (e.g. leech) with a more closed active site[85,86]. The wide active site was previously linked to a wide substrate specificity, showing that GH33 can cleave terminal sialic acids bound to glycans in varying bonds (e.g., α-2,3, α-2,6, α-2,8). The ability to acquire sialic acid from a variety of sources is beneficial for survival in the competitive ecological environment of the infant gut and it was previously shown that sialic acid drives changes in the gut microbiota composition and pathogen colonization[60,87].

Understanding how common infant gut microbes utilize sialylated HMOs and how the infant gut microbiome composition is modulated by these sugars will enable us to make informed decisions about formula supplementation, ultimately improving infant feeding and overall health. In the hypothetical matrix of which carbon sources can be utilized by which bacteria, there are still many unknowns. While these experiments are laborious, genetic manipulation tools tackling one pathway at a time are critical for gaining a mechanistic

understanding of the biological pathways accounting for carbon utilization in bacteria.

## Methods

In this study, we isolated strains from a total of 10 breastfed-infants stool samples. Mothers of all infants have agreed to participate in our study, which was approved by the Hebrew University's Institutional Review Board (IRB) (approval number 20042021), and signed our consent forms.

### Statistics and reproducibility

The previously published[71] differential expression analysis results, re-analyzed here, include fold-change values and adjusted *p*-values that were used to generate Fig. 1A, B, and the statistical tests used are described in the relevant section below. For all growth curve experiments, the experiments were performed using two biological replicates, in triplicates (*n* = 6), and were repeated twice. The statistical tests used for comparison between growth curves are described in the relevant section below. For genetic manipulation experiments, multiple colonies (20–100) were screened for the desired deletion using PCR and three mutants were identified. Figure 2C (left panel) shows four representative colonies for which deletion failed, and one representative mutant colony for which genetic manipulation was successful. Stool samples collected from breastfed infants were only used for strain isolation, and no other characterization of the samples was performed. No comparisons were made across samples or between samples collected for this study and other cohorts, and no sex, gender, or age analysis is relevant. The Investigators were blinded with respect to what stool samples they were isolating strains from. Predetermined sample size calculations or randomization considerations were less relevant here for these reasons.

### Microbial strains and growth conditions

The *B. uniformis* CL03T12C37 strain was kindly provided by the Geva-Zatorsky lab, Rappaport Technion Integrated Cancer Center, Israel. Anaerobes were cultured at 37 °C in an anaerobic chamber (COY) that was maintained at >2.5% $H_2$ by flushing with a gas mix containing 20% $H_2$, 5% $CO_2$, and 75% $N_2$. *Bacteroides* strains were routinely grown in supplemented Brain Heart Infusion (sBHI) media, supplemented with 50 ml/L fetal bovine serum (FBS; Sigma, F2442), 10 ml/L trace vitamins (ATCC® MD-VS™), 10 ml/L trace minerals (ATCC® MD-TMS™), 10 ml/L vitamin K1 and hemin (BBL, 212354), 1 g/L D-(+)-Cellobiose (Alfa Aesar, 528507), 1 g/L D-(+)-Maltose (Caisson, 6363537), 1 g/L D-(+)-Fructose (Sigma Aldrich, 1286504), and 0.5 g/L L-Cysteine (Acros Organics, 52904). For growth curve experiments, *Bacteroides* were grown in a *Bacteroides*-specific minimal media (MM) previously described[50], supplemented with 0.5% (w/v) carbon source. *E. coli* S17-1 lambda pir, used for conjugation, was grown in Luria Bertoni (LB) media. When appropriate, antibiotics were used at the following concentrations: 100 μg/mL ampicillin (Amp; Enco, 14417), 25 μg/mL erythromycin (Erm; Enco, 16486) and 200 μg/mL gentamicin (Gent; Enco, 1405-41-0). We used 100 ng/mL anhydrotetracycline (aTc; Enco, 10009542) for the induction of CRISPR-FnCpf1.

### Strain isolation

*B. dorei*, *B. vulgatus* and *B. stercoris* strains were isolated from breast-feeding infant stool samples as previously described[71]. Briefly, samples were transferred to −80 °C within 8 h. DNA was extracted using the DNeasy PowerSoil Pro Kit (QIAGEN) and used as a template for quantitative PCR (qPCR) reactions, with both control general 16S primers and *Bacteroides*-specific primers. Samples that passed the initial qPCR screening were plated on blood agar plates and then on selective bile esculin agar (BEA) plates and incubated anaerobically at 37 °C for 48 h. Single colonies were picked from the BEA plate, inoculated in liquid

sBHI and the bacterial identification to the species level was performed using Sanger sequencing of the 16S region.

### Growth curves

For each growth curve experiment, the strain of choice was plated on a sBHI agar plate that was incubated for 48 h at 37 °C. Two single colonies (biological replicates) were transferred to liquid sBHI media for overnight incubation. These cultures were then normalized to an optical density (OD600) of 1 and diluted 1:100 in MM (described above) with HMOs, lactose (positive control), or water (no carbon control) as the sole carbon source. The individual HMOs were received as a donation from DSM Nutritional Products Ltd (previously known as Glycom). All experiments were conducted with two biological replicates and three technical replicates per carbon source, under anaerobic conditions. Growth (OD600) was monitored every 30 min for 60 h, at 37 °C, using the Epoch2 microplate spectrometer (Agilent) with 96-well Costar culture plate (Corning, 3370). The optical density data was analyzed using R version 4.2.3[88]. Comparisons between growth curve graphs were performed using a paired two-sided *t*-test at two timepoints: for *B. dorei*, at one and 24 hours; for *B. uniformis* and *B. stercoris*, at one and 50 h, as they grow relatively slower. The *p*-values for the latter timepoints are reported in the main figures (Figs. 2D, 3A) and the full statistical data at both timepoints, including the mean difference between the OD600 values of the growth curves, are reported in Supplementary Table 1. A mean difference >0.1 with a *p*-value < 0.01 was considered statistically significant.

### CAZy enzymes per genome analysis

To calculate the average amount of sialidase and fucosidase enzymes encoded in the genome of *B. dorei* and *B. vulgatus*, we downloaded the CAZy database[53] and counted the number of unique genes annotated as GH33 (sialidase) or GH29 and GH95 (fucosidases) in these species. We then divided this result by 28, the sum of 15 *B. dorei* and 13 *B. vulgatus* genomes deposited in CAZy.

### RNAseq

Pure cultures of *B. dorei* growing on various carbon sources (in replicates) were harvested at log phase (OD600 ~ 0.7) using the Direct-zol™ RNA Miniprep Plus kit (Zymo Research, R2071). Agilent 2100 Bioanalyzer (Agilent Technologies) was employed for RNA quality control. Samples were processed according to a previously described protocol[89] and sequenced in two separate pools: HMOs (2'-FL, DFL, 3'-SL, 6'-SL, LNT, LNnT) and glucose. Single-end 75 bp sequencing was performed on a NextSeq device and the data was deposited in the National Center for Biotechnology Information (NCBI) repository, accession numbers SRS11934615 (HMOs) and SRS11934616 (glucose)[71].

The raw sequencing data were further filtered by trim_galore (https://github.com/FelixKrueger/TrimGalore) and classification of the *B. dorei* isolate to the strain level was achieved by alignment of reads using BLAST[90]. Genome mapping to *B. dorei* DSM 17855 genome (GenBank: CP046176.1) was performed by Bowtie2[91] and differential expression analysis between HMOs and glucose was carried out using featureCounts[92] and DEseq2[93]. DEseq2 uses the Wald test for differential expression analysis and the Benjamini and Hochberg method for multiple hypothesis correction. Significantly up-regulated genes were selected based on the *p*_adjusted <0.05 and log2 fold-change >1 parameters. Gene annotation of glycoside hydrolases (GHs) was downloaded from the CAZy (Carbohydrate Active enZYmes)[53] database.

### Conjugation

All plasmids (shuttle plasmids derivatives and pNBU2-ermGb derivatives) were introduced into *Bacteroides* via conjugation with *E. coli* S17-1, which harbors the conjugative machinery integrated onto chromosome[94]. To perform the conjugation, *E. coli* S17-1 and

*Bacteroides* strains were grown until the early stationary phase, to an OD600 that approximately equals to 0.2–0.3. Next, cells were combined at a ratio of 1:5 (donor:recipient, v/v) and centrifuged at 5000×*g* for 15–20 min. The mixed cell pellet was resuspended in 100uL sBHI media and spotted onto a sBHI agar plate for a 20–24 h aerobic incubation at 37 °C. Next, we scraped the cell mass from the plate, resuspended it into 1 mL sBHI media, and performed 10-fold serial dilutions, up to ×10$^5$. 100 μL of each serially diluted culture were plated on sBHI plates containing Gent and Erm and incubated for 2 days anaerobically, to select for transconjugants, which are *Bacteroides* cells that have accepted the plasmid into the cell.

## Plasmid construction

All strains and plasmids used in this study are listed in Supplementary Table 2 and the primers and genetic tools sequences that were used to construct them are listed in Supplementary Table 3. The shuttle plasmid for gene deletion, p_KO, is an all-in-one plasmid that encodes a Cas protein (FnCpf1), a custom single guide RNA (sgRNA), a repair template, and genetic elements for plasmid replication and conjugation[69]. The plasmid pB036 from Zheng et al. 2022 was kindly provided to us by the Lei Dai lab, Institute of Synthetic Biology, Chinese Academy of Sciences (Shenzhen). The plasmid pB036 was used as a template, which was linearized while replacing the gRNA, then assembled with the right and left flanking regions from *B. dorei* using Gibson Assembly (NEB, E2611) to create the circular p_KO. The gRNA was designed using CHOPCHOP[95].

Complementation plasmids were based on the previously described pNBU2-ermGb plasmid family[62]. pNBU2_erm_P5E1[67] was built from pNBU2_erm_SIE1 (Addgene, #136356), which was linearized excluding the SIE1 cassette, then assembled with a synthetically synthesized sequence of the P_BfP5E1 promoter and ribosome binding site (IDT, Syntezza Bioscience) using Gibson Assembly. Genes were cloned by Gibson Assembly downstream to P_BfP5E1, in the SalI (NEB, R3138) and NcoI (NEB, M032) digested pNBU2_erm_P5E1. p_compFull, containing the four knocked-out genes under constitutive expression from P_BfP5E1, was used as a template for partial complementation experiments. Primers were designed to linearize p_compFull excluding portions of it, then the linearized PCR product was circularized by T4 DNA ligase (NEB, M0202). The ligated plasmids (p_compPartial1/2/3) were sequenced by Nanopore sequencing for verification and their structure is depicted in Fig. 3A.

## Nanopore sequencing of isolates

The CRISPR-Cas12a genetic manipulation system[69] requires the target strain to be sensitive to erythromycin, but the *B. dorei* isolate on which RNA-seq was performed was resistant. We isolated an additional *B. dorei* strain from a breastfeeding infant stool sample, which is erythromycin-susceptible. To validate the compatibility of the additional *B. dorei* isolate for genetic manipulation, we performed whole genome sequencing of both isolates. High-molecular-weight DNA was extracted from pure cultures of the isolates using the MagAttract HMW DNA Kit (Qiagen, 67563), and then processed using the Rapid Sequencing kit (ONT, SQK-RBK004) for library preparation. Whole genome nanopore sequencing was performed on a Minion device, using a R9.4.1 flow cell. Basecalling was performed using Guppy version 6.1.2 (ONT), quality assessment was performed using Nanoplot version 1.33.1[96], read filtering and trimming was performed using Porechop version 0.2.4 (https://github.com/rrwick/Porechop) and assembly was performed using Flye version 2.9[97]. The final assembled genomes of both isolates were deposited in the NCBI database under GenBank assembly accessions GCA_030122685.1 (*B. dorei* 1) and GCA_030122665.1 (*B. dorei* 2). The average nucleotide identity (ANI) in Supplementary Fig. 4B was calculated using the OrthoANIu ANI calculator[98]. The synteny map in Supplementary Fig. 4B, comparing the genomes of the two *B. dorei* isolates to the type strain *B. dorei* DSM

17855, was constructed using the gggenomes package (https://github.com/thackl/gggenomes) under R version 4.2.3[88].

## Markerless gene deletion *Bacteroides*

The plasmid for gene deletion, p_KO, was first transformed into *E. coli* S17-1 and then introduced into *B. dorei* using conjugation. Transconjugants were selected on sBHI + Gent + Erm as described above, then the Cas protein (FnCpf1) was induced to activate the gene editing process. Transconjugants were incubated overnight in sBHI + Gent + Erm + aTC, then plated on sBHI + aTC agar plates and incubated for two days until colonies were observed. Colonies were PCR-screened for the correct deletion using primers matching the genome of *B. dorei*, before and after the expected deletion site. The identified mutants were further verified using Sanger sequencing of the deletion region in *B. dorei*'s genome.

Plotting of PUL33 and the deleted area was performed using clinker[99] version 0.0.27.

## Complementation assays

The pNBU2-ermGb plasmid derivatives harbor the IntN2 tyrosine integrase, which mediates the recombination between the attN site on the plasmid and one of the two attBT sites at the 3' end of the tRNASer genes (GKD17_04220 and GKD17_06425) on *B. dorei*'s chromosome[100]. Thus, the pNBU2-ermGb plasmid derivatives were integrated into the chromosome following conjugation.

## Homology search of PUL33

Prior to the conjugation of *B. uniformis* CL03T12C37 and our *B. stercoris* isolate with genes from PUL33, in addition to phenotypically showing that they cannot grow on sialylated HMOs in vitro, we verified the absence of PUL33 homologs in their genome. For the type strain *B. uniformis* CL03T12C37, the sequence is publically available (GenBank accession number CP072255.1) and also deposited in the PULDB database[74], with one GH33 gene encoded in PUL16, genes INE75_01450 to INE75_01456. The genomic context of GH33 and the gene structure of PUL16 in *B. uniformis* varies from that of *B. dorei*'s PUL33, with the *B. uniformis* cluster encoding SusC, SusD, GH33, an aldose 1-epimerase gene and three additional genes with unknown functions, lacking GH2 or GH20 genes present in *B. dorei*'s PUL33. Thus we concluded that the complementation of *B. uniformis* CL03T12C37 with genes from *B. dorei*'s PUL33 could provide capabilities which are non-overlapping with the existing repertoire of PULs encoded in *B. uniformis*. For *B. stercoris*, only one of six genomes deposited in PULDB encoded a GH33 gene[74], so combined with the experiment inability of our isolate to utilize sialylated HMOs (Supplementary Fig. 6), we concluded that it doesn't encode a cluster homologous to *B. dorei*'s PUL33.

## Real-time qPCR (RT-qPCR)

Pure cultures of *B. dorei* and *B. dorei* Δ5k-PUL33::p_compPartial1/2/3 growing on lactose, 3'-SL and 6'-SL (in replicates) were harvested at log phase (OD ~ 0.7) using the Direct-zol™ RNA Miniprep Plus kit (Zymo Research, R2071). Reverse transcription of the total RNA samples to cDNA was performed using SuperScript™ III Reverse Transcriptase (Invitrogen, 18080093). RT-qPCR reaction volumes were 4 μL 0.1 ng/μL DNA, 0.5 μL 10 μM forward primer (500 nM), 0.5 μL 10 μM reverse primer (500 nM), 5 μL iTaq Universal SYBR Green Supermix (BioRad, 1725124). For SusC only, the primer concentration was found to be optimal at 100 nM. The amplification program consisted of (1) 95 °C for 30 s, (2) 95 °C for 10 s (3) 60 °C for 30 s (4) repeat 2–3 for 39 times. The fluorescent products were detected at the last step of each cycle using a CFX96 instrument (BioRad). Primer efficiencies were optimized and calculated using a standard curve and relative changes in gene expression were calculated compared to the housekeeping gene rpsL[101], with the efficiency-corrected ΔCq method[102].

## Analysis of GH33 sequences

The phylogenetic tree of enzymes from the GH33 family in Supplementary Fig 8 was constructed using the SACCHARIS pipeline[76], for randomly selected GH33 protein sequences deposited to the CAZy database in addition to *B. dorei*'s GH33 sialidase discussed in this manuscript and the *B. thetaiotaomicron* VPI 5482 crystallized GH33 protein[59] (GenBank accession AAO75562.1; PDB code 4BBW). Alignment of protein sequences in Supp. Fig 9 was performed using Clustal Omega[103], which utilizes the HHalign algorithm[104] for calculating distances. The root mean square deviation (RMSD) values were calculated after superimposing the atomic coordinates of the proteins, using PyMOL's (version 2.5.4[105]) 'align' function. Since the GH33 protein 3D structure of our *B. dorei* isolate and *B. vulgatus* ATCC 8482 (GenBank accession UBD82890.1) is predicted and not crystallized, we excluded residues 1–23 and 545–546, for which the predicted local distance difference test value (pLDDT) was lower than 70, indicating a low per-residue confidence score. On this account, the first 23 residues were also excluded from the crystallized *B. thetaiotaomicron* VPI 5482 GH33 protein sequence (its length is 544 so residues 545–546 were not relevant). In total, 521 residues from each of the three proteins were aligned for the RMSD calculation.

## Reporting summary

Further information on research design is available in the Nature Portfolio Reporting Summary linked to this article.

## Data availability

The RNA-seq data used in this study is deposited in the NCBI database under SRA accession codes SRS11934615 (HMOs) and SRS11934616 (glucose). The previously published differential expression analysis results, that were re-analyzed here, are available under DOI 10.3389/fcimb.2022.854122[71]. The Nanopore sequencing whole genome assembly data for *B. dorei* isolates generated in this study is deposited in the NCBI database under GenBank assembly accessions GCA_030122685.1 (*B. dorei* 1) and GCA_030122665.1 (*B. dorei* 2). Source data are provided with this paper.

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

## Acknowledgements

We thank the Lei Dai lab and specifically Liggang Zheng, for plasmids and assistance with setting up the genetic manipulation system; The Goodman lab and Natasha A. Bencivenga-Barry, for the pNBU_erm plasmid; The DSM Nutritional Products Ltd (previously known as Glycom) for the HMO donation; Naama Geva-Zatorsky and her lab members for the *B. uniformis* CL03T12C37 strain; Illan Rosenshine's lab for *E. coli* S17-1 lambda pir strain; Seth Rakoff-Nahoum for fruitful discussions and experimental feedback; Dina Schneidman for guidance regarding the protein structure analysis. This project, S.K. and M.Y. were supported by an Israeli Ministry of Health grant (3-18355), and an Israel Science Foundation grant (2660/18). A.F. is supported by The Abisch-Frenkel Faculty Development Lectureship. M.Y. is the Rosalind, Paul and Robin Berlin Faculty Development Chair in Perinatal Research, also supported by the Azrieli Foundation.

## Author contributions

S.K. established the experimental system & performed experiments and analysis. D.E. and S.S. assisted in establishing the experimental system. A.F. guided the genetic manipulation work. M.Y. guided the work and the computational analysis. S.K. and M.Y. wrote the manuscript. All authors contributed to the article and approved the submitted version.

## Competing interests

The authors declare no competing interests.
