## [Peer Review File · Nature Communications]

REVIEWER COMMENTS

Reviewer #1 (Remarks to the Author):

The manuscript by Kijner et al investigates the utilization of HMOs by the infant gut symbiont *Bacteroides dorei*. The authors used a *B. dorei* strain isolated from infant stool in previous published work. Here, they carried out RNAseq analysis of *B. dorei* grown on different HMOs, and identified a PUL for utilization of sialylated HMOs, which they validated in vitro using genetic editing. The work is sound but, as it stands, is limited in scope and novelty. Beyond the use of CRISPR editing tools to make *Bacteroides* mutant, the main outcome of this work is that GH33 is involved in sialylated HMOs degradation by *B. dorei*, which is to be expected from work in the field.

General comments

- The claims that ‘the utilisation of HMOs by *Bacteroides* is completely unknown’ (as stated in the abstract and further in the text) should be revised in view of the many papers (some of them referred to by the authors in the Introduction L50-54) reporting that *Bacteroides* strains can utilize HMOs through regulation of PULs (using growth assays, bioinformatics and transcriptomics approaches, and the characterization of *Bacteroides* genetic mutants in HMO-specific loci)
- In the Introduction, the authors seem to infer novelty by the fact that they used individual HMOs (commercially available) rather than pool of HMOs to investigate HMO degradation strategy by *Bacteroides* (L 63-65), which is quite thin (as compared to the broad claim of the abstract).
- Some references in the field are lacking in the Introduction that may be relevant to this work. Eg.
 - Park KH, Kim MG, Ahn HJ, Lee DH, Kim JH, Kim YW, Woo EJ. Structural and biochemical characterization of the broad substrate specificity of *Bacteroides thetaiotaomicron* commensal sialidase. *Biochim Biophys Acta*. 2013 Aug;1834(8):1510-9.
 - Moya-González EM, Peña-Gil N, Rubio-Del-Campo A, Coll-Marqués JM, Gozalbo-Rovira R, Monedero V, Rodríguez-Díaz J, Yebra MJ. Infant Gut Microbial Metagenome Mining of α -L-Fucosidases with Activity on Fucosylated Human Milk Oligosaccharides and Glycoconjugates. *Microbiol Spectr*. 2022 Aug 31;10(4):e0177522.
- It is not clear why the study only focused on sialylated HMO utilisation pathway while other HMOs were tested.
- The title and abstract should reflect the focus of the work.
- The results are quite narrow in scope, and it would be interesting to follow up these observations by studies investigating the structure function relationships of *B. dorei* GH33 sialidases or the biological role of these enzymes in relation to infant gut adaptation.

Specific comments

Introduction

- L65- Reference missing
- L 75 'Here we have isolated Bacteroides dorei stains from breastfeeding stool samples and examined...' This sentence is misleading as it infers that the isolation is part of this study, while the authors use strains previously isolated in a previously published study.

Results

- Could the authors discuss the RNAseq results from other HMOs tested in this work?
- L.100, what is meant by 'consistent' upregulation pattern (in which way is it consistent? between replicates? between condition tested?).
- L.115-116 sentence would need to be reformulated.
- When testing other Bacteroides species, why not using complementation with the plasmids lacking GH20, GH33 and SusCD, which would provide more insightful information than the use of the p-compFull plasmid?

Discussion

- Description of transporters in L. 189-197- is a bit naïve and would need to be substantiated by bioinformatics analyses and/or references to relevant literature on sialic acid transporters in bacteria.
- It is not clear how the work reported in the manuscript link to the last paragraph on potential biological roles of sialylated HMOs in infant health.

Reviewer #2 (Remarks to the Author):

Summary

Human milk oligosaccharides (HMOs) are a fascinating group of chemicals: their primary purpose appears to be the nourishment of the gut microbiota of newborns, which in turn can have important long-term consequences for health, lasting well into adulthood. As such it is important to understand how different gut microbes catabolize the different HMOs and how such catabolism leads to establishment of a functional microbiota. Most research thus far has focused on HMO catabolism in Bifidobacteria. Other genera are known to have this ability as well, but through mechanisms that are poorly characterized. In this manuscript, the authors describe the identification of a polysaccharide

utilization locus (PUL) in *Bacteroides dorei* that is required for growth on sialylated HMOs. The authors initially identified the PUL as transcriptional upregulated during growth in the presence of HMOs and further demonstrate through knockout and complementation studies that genes within this cluster are important for catabolism of sialylated HMOs. Importantly, heterologous expression of four genes from this PUL in other *Bacteroides* strains confers some ability to catabolize sialylated HMOs. The findings are timely and important, and the manuscript overall is clear. That said, I feel that the manuscript lacks some essential context, that some of the findings in the manuscript are overstated, and that some potential alternative explanations are not examined.

Major comments

1. The authors have previously published this paper <https://www.frontiersin.org/articles/10.3389/fcimb.2022.854122/full> , which is only mentioned in passing even though it is highly related. The authors should present and discuss their findings in the context of their previous work and make it clear what aspects of this manuscript are indeed new.

a. The RNA-seq data reported in this manuscript appear to have been published in the previous paper. The text currently reads as if the authors had performed new RNA-seq studies, but it appears that the authors re-analyzed their previous data. The authors must make it clear that this is the case.

b. The authors state in the introduction that they examined the growth kinetics of different *B. dorei* strains in the presence of six HMOs. Again these data are not presented in this manuscript and appear to be part of the previous paper. Please either include these data, if they were newly acquired, or edit the text.

2. The authors discuss RNA-seq data of *Bacteroides dorei* grown in the presence of six different HMOs but then focus their analysis on sialylated HMOs. Although this focus leads them to a candidate gene required for use of those sialylated HMOs, the focus is not rationalized otherwise. Were other PULs or genes differentially expressed during growth in the presence of other HMOs? If so, what motivated the focus on sialylated HMOs and PUL32?

a. As a minor point, the threshold that only genes with $p < 1.5 \cdot 10^{-10}$ are considered “significantly” upregulated seems arbitrary. Please provide rationale for this threshold. Alternatively, use a more established threshold (e.g. 0.01 or 0.001) and include a discussion of additional genes that now match this threshold.

3. For antibiotic sensitivity reasons, the authors used different isolates of *B. dorei* for the RNA-seq experiment and the genetic manipulation experiments. The authors outline their reasoning in the Methods but should also mention it in the Results section. Additionally, the authors should validate their main finding from Figure 1, namely that genes in PUL32 are upregulated upon growth on sialylated HMOs, on the isolate used for genetic manipulation, for example by RT-qPCR.

4. The authors' claims that the four genes from PUL32 are necessary and sufficient for breakdown of sialylated HMOs are only partially supported by the evidence and overstated as currently written. The main lines of evidence are the complementation experiments in a) their *B. dorei* strain with knockout of the four genes and b) additional *Bacteroides* strains. Although these experiments provide some support for the claim of sufficiency, both have many alternative interpretations and as such are inconclusive and leave several questions open.

a. The authors deleted a 5 kb region encompassing GH20, GH33, and SusCD. What is the consequence of this genetic manipulation for expression of the other genes in PUL32? Are these genes still upregulated by sialylated HMOs in the knockout/complementation strains? If they are still expressed, this would weaken the argument that the deleted genes are sufficient for use of sialylated HMOs.

b. The authors should also provide further rationale for deleting the 5 kb region encompassing the four genes. Why did they choose not to delete the entire cluster?

c. On a technical note, the complemented *B. dorei* strains grow to a lower final OD than the wild-type strain. Could the authors comment on potential explanations?

d. The second line of evidence for sufficiency is that expression of the four genes from PUL32 allows *B. uniformis* and *B. stercoris* to grow in the presence of sialylated HMOs. That said, it is not clear how well the strains are growing. The authors should include growth traces in the presence of a preferred carbon source for each strain and/or lactose. In addition, the authors should note if these strains encode any homologs of the remaining genes in PUL32 or contain any PULs that are similar to PUL32.

e. The authors further claim that GH33 is the main gene that is required for use of sialylated HMOs. To further probe sufficiency, the authors should complement this gene on its own, in their *B. dorei* knockout strain as well as in the additional *Bacteroides* strains.

f. In their previous paper, the authors described that some *B. vulgatus* isolates are also able to grow on sialylated HMOs and that the *B. vulgatus* isolates also contain a GH33 that is upregulated in the presence of sialylated HMOs. Is this GH33 similar to the GH33 the authors focus on here and is it embedded in a PUL similar to PUL32 in these *B. dorei* isolates? This could provide further evidence that this GH33 is the important gene for breakdown of sialylated HMOs.

I realize this is a substantial amount of work, and the authors need not conduct all of it. For example, it may be acceptable to address points a and d. As a complement, it would be important to tone down claims of sufficiency for sialylated HMO use in the manuscript.

5. The authors state that GH33-PUL8 is insufficient to rescue growth on sialylated sugars and hypothesize that the gene has distinct functions from GH33-PUL32. It is possible that the gene has similar function but is not expressed under the growth conditions used in this study. The authors show that GH33-PUL8 is not significantly upregulated upon growth in sialylated HMOs but it is not clear what the absolute expression levels of the gene are. Is the GH33-PUL8 gene expressed at appreciable levels under the conditions used here?

6. The introduction makes the utilization of HMOs by *Bacteroides* seem unstudied, while also introducing research that has already shown other *Bacteroides* species as known HMO consumers. I would recommend toning down claims of complete lack of knowledge into this process. Additional points:

a. Line 53: “Taken together, all these suggest that *Bacteroides* species are the immediate next candidates for HMS utilization” hints at a hierarchy between genera that is not clear, and the phrasing undersells the previous work identifying *Bacteroides* as HMO consumers – it may be better to change the wording to emphasize that *Bacteroides* species are known HMO utilizers, but that many of the mechanisms remain unknown.

b. Line 63-65: Please cite again references to which this statement refers.

Minor Comments

1. The authors validate their knockout strain using PCR with primers flanking the region targeted for knockout. Although this is a good approach in general, it is fraught for large deletions: it is impossible to detect if there are cells containing the wild-type genotype within the population, because the PCR will strongly favor the smaller product. I would recommend supplementing these data with PCRs with pairs of primers that uniquely recognize the junctions in the different genotypes. For example, one primer pair could be designed with one primer outside of the targeted region and one primer inside the targeted region, and demonstrating that this primer pair produces a product only in the wild-type strain.

2. The pink colors used to indicate 3-SL and 6-SL in the growth curves throughout the paper are very similar and difficult to differentiate. I suggest using two colors with a stronger contrast.

3. Lines 80 and 144: I would recommend that the authors be careful with their use of “essential”, as many readers will interpret that as “essential for growth” rather than “essential for HMO utilization”. Alternative wording could include “important” or “necessary”.

4. Line 67: Change “parts” to “tools”.

5. Line 83: Delete “a”
6. Line 108: please note fold changes instead of log₂ fold changes.
7. Line 131: tone down claim of “fully restored”, as the complemented strains appear to have defects in growth compared to wild-type.
8. Line 157: Please include the data in the supplemental figures.
9. Especially in the discussion, please refer to the strains by their genotypes rather than as p_compPartial3 etc, e.g. “B. dorei strain lacking/expressing X”.
10. Figure 2A: please include the Cas12 gene in the schematic of the plasmid.
11. Figure 2D: It is unclear how the statistical comparisons of the WT to Δ5k strains, and Δ5k to p_comp strains was performed. The methods states a paired t-test at two timepoints, but only provides a single p-value per carbon source.
12. Supp. Fig. 3: It is not clear what the difference between the two lactose/no carbon panels is. Additionally, it looks like mean lines are plotted in a slightly lighter shade along with the individual samples which makes the data hard to read.

We would like to thank the reviewers for their thorough and insightful feedback. Following your suggestions, we have made significant modifications to the manuscript. Before addressing your comments specifically, we would like to point out that since our submission, an update to the CAZy database has been released, reclassifying the gene GKD17_02665, referred to as GH33-PUL8 in our manuscript. It was previously annotated as GH33 in PUL8, however in the new version PUL8 was split into two distinct PULs (termed now PUL8 and PUL9), excluding the GKD17_02665 gene. This split of PULs affected the numbering of subsequent PULs, meaning that PUL32 is currently PUL33. We have corrected naming across the manuscript and figures, however addressed PUL33 as PUL32 in this document.

Additionally, we attempted to obtain additional sialylated HMOs for a wider scope, however this effort will probably take a long while as changes have been made to the DSM Nutritional Products Ltd donation program terms, and the legal teams on both sides are currently negotiating.

Reviewer #1:

The manuscript by Kijner et al investigates the utilization of HMOs by the infant gut symbiont *Bacteroides dorei*. The authors used a *B. dorei* strain isolated from infant stool in previous published work. Here, they carried out RNAseq analysis of *B. dorei* grown on different HMOs, and identified a PUL for utilization of sialylated HMOs, which they validated in vitro using genetic editing. The work is sound but, as it stands, is limited in scope and novelty. Beyond the use of CRISPR editing tools to make *Bacteroides* mutant, the main outcome of this work is that GH33 is involved in sialylated HMOs degradation by *B. dorei*, which is to be expected from work in the field.

We thank the reviewer for the detailed response and comments. We have carefully read all the issues, and addressed them by adapting the text and/or adding experimental results to the manuscript. We thank the reviewer for assisting us in improving this new and revised version.

General comments

1. The claims that ‘the utilisation of HMOs by Bacteroides is completely unknown’ (as stated in the abstract and further in the text) should be revised in view of the many papers (some of them referred to by the authors in the Introduction L50-54) reporting that Bacteroides strains can utilize HMOs through regulation of PULs (using growth assays, bioinformatics and transcriptomics approaches, and the characterization of Bacteroides genetic mutants in HMO-specific loci)

We thank the reviewer for pointing this out. We have modified the text to address these issues (deleted the “completely unknown” phrase from the abstract). However, it is important to note that there is no existing manuscript that characterizes *Bacteroides* genetic mutants in HMO-specific loci. There are works examining specific sugars (i.e., Feng et al. 2022, ref 70), yet none looking at HMOs. We elaborated more on the status of current knowledge and previous research regarding HMO utilization by *Bacteroides* in the introduction L62-67: “Some of the mechanisms have been further identified, for example, that *B. thetaiotaomicron* and *B. fragilis* utilize both host mucin-O-glycans and HMOs using a similar set of upregulated genes⁵⁶. For fucosylated HMOs, fucosidases from *Bacteroides* were profiled and showed versatility in efficiency and substrate bond specificity⁵⁸. Finally, for sialylated sugars, one of *B. thetaiotaomicron*’s GH33 sialidases was crystallized and characterized⁵⁹, unraveling a wide binding groove accounting for a broad substrate specificity, including various sialylated HMOs⁶⁰.”

2. In the Introduction, the authors seem to infer novelty by the fact that they used individual HMOs (commercially available) rather than pool of HMOs to investigate HMO degradation strategy by Bacteroides (L 63-65), which is quite thin (as compared to the broad claim of the abstract).

We re-wrote these sections of the introduction, and stated this point more clearly. We now specify the type of individual HMO studies that were performed (L62-67 as in the previous comment), and state the need for additional types of individual HMOs to be studied in this context, in L68-69: “While these pioneering works lay the foundation for understanding HMO utilization by *Bacteroides* species, further research is needed to extend the

mechanistic characterization to additional infant gut commensal strains, and specific types of HMOs.”

3. Some references in the field are lacking in the Introduction that may be relevant to this work. Eg.

- Park KH, Kim MG, Ahn HJ, Lee DH, Kim JH, Kim YW, Woo EJ. Structural and biochemical characterization of the broad substrate specificity of *Bacteroides thetaiotaomicron* commensal sialidase. *Biochim Biophys Acta*. 2013 Aug;1834(8):1510-9.

- Moya-González EM, Peña-Gil N, Rubio-Del-Campo A, Coll-Marqués JM, Gozalbo-Rovira R, Monedero V, Rodríguez-Díaz J, Yebra MJ. Infant Gut Microbial Metagenome Mining of α -L-Fucosidases with Activity on Fucosylated Human Milk Oligosaccharides and Glycoconjugates. *Microbiol Spectr*. 2022 Aug 31;10(4):e0177522.

Thank you very much for pointing these out, we added them to the introduction (references number 58 & 59).

4. It is not clear why the study only focused on sialylated HMO utilisation pathway while other HMOs were tested.

Thank you for raising this point. We added a paragraph in the Results to address this point, L112-119:

“Hereafter, we chose to focus on *B. dorei* isolates growing on sialylated HMOs due to a number of reasons. First, sialylated HMOs contribute to infant health, and their effect is mediated by the gut microbiome (see **Introduction**). Second, the consistent growth dynamics on the two tested sialylated HMOs (3'-SL & 6'-SL), together with the unique terminal residue of sialic acid suggest a specialized pathway or mechanism to be discovered, which is specific for sialylated sugar structures. Lastly, *B. dorei*'s growth on sialylated HMOs is overall comparable to glucose and lactose, especially in contrast to the poor growth on fucosylated structures, suggesting that sialylated HMOs serve as an effective carbon source for the species *B. dorei*.”

5. The title and abstract should reflect the focus of the work.

We changed the title to be more specific to sialylated HMOs, and toned-down the statements in the abstract. The new title is: “Identification of a novel sialylated Human Milk Oligosaccharides utilization cluster in the infant gut commensal *Bacteroides dorei*”.

6. The results are quite narrow in scope, and it would be interesting to follow up these observations by studies investigating the structure function relationships of *B. dorei* GH33 sialidases or the biological role of these enzymes in relation to infant gut adaptation.

We agree that additional follow up studies of the structure and function of these enzymes would be extremely interesting and useful, yet they are outside the scope of this specific paper, and require a different set of expertise. We added a section to the Results discussing the predicted structure and derived function of the GH33 enzyme in *B. dorei* (Predicted structure of *B. dorei*'s GH33 suggests a broad substrate specificity, L219-242).

Specific comments

Introduction-

1. L65- Reference missing

We have altered this sentence and referred to manuscripts that demonstrated HMO utilization by *Bacteroides* or further investigated the mechanistic basis, characterizing the structure and function of sialidases and fucosidases in *Bacteroides*. We elaborate more on the changes we have made to that part of the text in the response to your first general comment. The relevant references are 56-60 (L59-67): Marcobal et al. 2011, Salli et al. 2021, Moya-González et al. 2022, Park et al. 2013, Bell et al. 2023.

2. L 75 ‘Here we have isolated *Bacteroides dorei* strains from breastfeeding stool samples and examined...’ This sentence is misleading as it infers that the isolation is part of this study, while the authors use strains previously isolated in a previously published study. We thank the reviewer for raising this point. We have now substantially enhanced this Results section, clarifying the source of the *B. dorei* isolates investigated here (L92-93 “we started with two *B. dorei* strains that we recently isolated, which were shown to grow on various HMOs⁷¹”), and adding eight more isolates of *B. dorei* and *B. vulgatus* to the manuscript. In the previously published study we focused on establishing a system for the isolation of *Bacteroides* from stool samples. We demonstrated the efficiency of this system by isolating strains belonging to a variety of *Bacteroides* species from infant stool samples.

Following your comment, to improve the flow of the paper and make the distinction between the previous paper and this one clearer, we added a section to the beginning of the Results section. We first show the growth kinetics of 10 *B. dorei* and *B. vulgatus* (closest relatives, see ref 72, García-López et al. 2019) on the set of six HMOs (2'-FL, DFL, 3'-SL, 6'-SL, LNT, LNnT). Eight of these isolates are new and were not previously published, and it is now stated clearly, L93-97:

“Before we continued to mechanistically examine the HMO utilization process in these strains, we first explored the generality of this phenotype in additional natural isolates. Towards this end, we isolated two additional *B. dorei* strains and six *B. vulgatus* strains (which are closest taxonomically in the *Bacteroides* species tree⁷²) from breastfed-infant stool samples.”

Then, we zoom into one strain, show the transcriptional profiling results with the PUL32 upregulation, and move to another strain to perform genetic manipulation. We also verified that PUL32 is upregulated in the genetically-modifiable strain via RT-qPCR (see response to major comment 3 reviewer #2).

Results-

3. Could the authors discuss the RNAseq results from other HMOs tested in this work?

We chose to focus in this paper on the response to sialylated HMOs, as explained above (major comment 4). The data, however, is published, and additional labs that are specifically interested in additional HMOs can mimic this analysis on their preferred HMO family. When choosing the sialylated HMOs as the focus of this work, we saw that there is no significant single PUL that is strongly upregulated in fucosylated HMOs, thus, again,

we chose not to focus on it here, and it can serve as interesting follow ups by additional labs.

4. L.100, what is meant by 'consistent' upregulation pattern (in which way is it consistent? between replicates? between condition tested?).

Our intention with the word "consistent" was to emphasize that PUL32 is the only cluster for which all genes were upregulated on sialylated HMOs, whereas for other genomic clusters this is not the case. We corrected the text to be clearer, from "...none exhibited a consistent upregulation pattern similar to PUL32" to "Of the remaining 65 PULs in *B. dorei*'s genome⁷⁴, none were completely upregulated as PUL32 was, rather only some of their genes were upregulated" (L127-128).

5. L.115-116 sentence would need to be reformulated.

We re-wrote this sentence from "To test the essentiality of PUL32 in the utilization of sialylated HMOs, we wanted to examine *B. dorei*'s ability to grow on minimal media with 3'-SL and 6'-SL, without the specific genes in PUL32" to "In order to test if PUL32 is essential for the utilization of sialylated HMOs, we aimed to assess the growth of *B. dorei* on minimal media containing 3'-SL and 6'-SL, while lacking specific genes in PUL32."

6. When testing other *Bacteroides* species, why not using complementation with the plasmids lacking GH20, GH33 and SusCD, which would provide more insightful information than the use of the p-compFull plasmid?

Thank you for your insightful comment. We added partial complementation experiments in the additional *Bacteroides* strains to the Results section, L212-218:

"A similar pattern was observed for *B. uniformis* CL03T12C37, for which complementation with plasmids containing the GH33 gene (p_compPartial1/3/4) allowed modest growth on 3'-SL and 6'-SL, whereas complementation with the p_compPartial2 plasmid that lacks GH33 did not convey this ability (**Fig 3B,C**). However, for the *B. stercoris* isolate, none of the partial complementation plasmids were sufficient to drive significant growth on sialylated HMOs (**Supp. Fig 6**), suggesting additional functions are missing in the *B. stercoris* cellular environment to enable even a modest growth."

Discussion-

7. Description of transporters in L. 189-197- is a bit naïve and would need to be substantiated by bioinformatics analyses and/or references to relevant literature on sialic acid transporters in bacteria.

We altered the text to be more specific and to state that this concept needs further support. We added references to literature regarding sialic acid transport in bacteria, number 83 & 84.

8. It is not clear how the work reported in the manuscript link to the last paragraph on potential biological roles of sialylated HMOs in infant health.

In the original text we wanted to highlight the specific importance of sialylated HMOs within this very big group of diverse sugars. To address the reviewers point, we moved

this text to the introduction (L37-41), emphasizing sialylated HMOs from the beginning of the manuscript, and trimmed the last paragraph of the discussion accordingly.

Reviewer #2:

Human milk oligosaccharides (HMOs) are a fascinating group of chemicals: their primary purpose appears to be the nourishment of the gut microbiota of newborns, which in turn can have important long-term consequences for health, lasting well into adulthood. As such it is important to understand how different gut microbes catabolize the different HMOs and how such catabolism leads to establishment of a functional microbiota. Most research thus far has focused on HMO catabolism in *Bifidobacteria*. Other genera are known to have this ability as well, but through mechanisms that are poorly characterized. In this manuscript, the authors describe the identification of a polysaccharide utilization locus (PUL) in *Bacteroides dorei* that is required for growth on sialylated HMOs. The authors initially identified the PUL as transcriptional upregulated during growth in the presence of HMOs and further demonstrate through knockout and complementation studies that genes within this cluster are important for catabolism of sialylated HMOs. Importantly, heterologous expression of four genes from this PUL in other strains confers some ability to catabolize sialylated HMOs. The findings are timely and important, and the manuscript overall is clear. That said, I feel that the manuscript lacks some essential context, that some of the findings in the manuscript are overstated, and that some potential alternative explanations are not examined.

We thank the reviewer for the time and effort in their comments. We significantly revised the text, taking special consideration to not overstate our results. Full details appear below following each comment and suggestion.

Major comments

1. The authors have previously published this paper <https://www.frontiersin.org/articles/10.3389/fcimb.2022.854122/full> , which is only mentioned in passing even though it is highly related. The authors should present and discuss their findings in the context of their previous work and make it clear what aspects of this manuscript are indeed new.
 - a. The RNA-seq data reported in this manuscript appear to have been published in the previous paper. The text currently reads as if the authors had performed new RNA-seq studies, but it appears that the authors re-analyzed their previous data. The authors must make it clear that this is the case.

We thank the reviewer for this comment, and we altered to text to be clear about the source of the RNA-seq data, L122: "Here, we re-analyzed previously published⁷¹ RNA-sequencing data of one of our *B. dorei* isolates..."
 - b. The authors state in the introduction that they examined the growth kinetics of different *B. dorei* strains in the presence of six HMOs. Again these data are not presented in this manuscript and appear to be part of the previous paper. Please either include these data, if they were newly acquired, or edit the text.

We thank the reviewer for pointing this out. This was also brought up by Reviewer #1, we have substantially changed the text to clarify the results that are based on the published work vs. the new results from this work. For further details on the

revised text with regards to the previous paper, and also regarding the various isolated strains, please see response to “Specific comment 2 to Reviewer #1”.

2. The authors discuss RNA-seq data of *Bacteroides dorei* grown in the presence of six different HMOs but then focus their analysis on sialylated HMOs. Although this focus leads them to a candidate gene required for use of those sialylated HMOs, the focus is not rationalized otherwise. Were other PULs or genes differentially expressed during growth in the presence of other HMOs? If so, what motivated the focus on sialylated HMOs and PUL32?

This important point was also raised by Reviewer #1, and we added another paragraph explaining why we chose to focus on sialylated HMOs (L112-119), for full details please see response to “General comment 4 Reviewer #1”. From the RNA-seq data, other HMO types (fucosylated, neutral) did not have a full PUL that was upregulated when used as a single carbon source, whereas all genes of PUL32 were consistently upregulated on both 3'-SL and 6'-SL. Thus we chose to focus here on sialylated HMOs.

- a. As a minor point, the threshold that only genes with $p < 1.5 \cdot 10^{-10}$ are considered “significantly” upregulated seems arbitrary. Please provide rationale for this threshold. Alternatively, use a more established threshold (e.g. 0.01 or 0.001) and include a discussion of additional genes that now match this threshold.

We clarified the phrasing used in the text. We did use a threshold of $p < 0.001$ to classify a significant upregulation, and only mentioned the specific p-values found for that PUL in the text. We now revised it, and it is much clearer, L124-126: “The whole genome transcriptional profiling analysis revealed that a specific gene cluster called PUL32 is significantly upregulated when growing on sialylated HMOs compared to glucose ($\log_2(\text{fold-change}) > 1$, $\text{padj} < 0.001$; **Fig 1A,B**)”. Thank you for noticing this point.

3. For antibiotic sensitivity reasons, the authors used different isolates of *B. dorei* for the RNA-seq experiment and the genetic manipulation experiments. The authors outline their reasoning in the Methods but should also mention it in the Results section. Additionally, the authors should validate their main finding from Figure 1, namely that genes in PUL32 are upregulated upon growth on sialylated HMOs, on the isolate used for genetic manipulation, for example by RT-qPCR.

We thank the reviewer for pointing this out. We performed the RT-qPCR to show PUL32 upregulation (Supp Figure 4A), and also updated the text in the Results to be clear on this point (L151-154): “As the CRISPR-based system requires sensitivity to erythromycin, we implemented it on an erythromycin-sensitive *B. dorei* isolate from our strain collection, and confirmed by RT-qPCR that it also upregulates PUL32 upon growth on sialylated HMOs”.

4. The authors' claims that the four genes from PUL32 are necessary and sufficient for breakdown of sialylated HMOs are only partially supported by the evidence and overstated as currently written. The main lines of evidence are the complementation experiments in a) their *B. dorei* strain with knockout of the four genes and b) additional

Bacteroides strains. Although these experiments provide some support for the claim of sufficiency, both have many alternative interpretations and as such are inconclusive and leave several questions open.

- a. The authors deleted a 5 kb region encompassing GH20, GH33, and SusCD. What is the consequence of this genetic manipulation for expression of the other genes in PUL32? Are these genes still upregulated by sialylated HMOs in the knockout/complementation strains? If they are still expressed, this would weaken the argument that the deleted genes are sufficient for use of sialylated HMOs. If the additional genes in PUL32 are not expressed under sialylated HMOs growth, it will suggest that we have harmed the regulation of this cluster. However, even if these genes are still expressed, indicating that the regulation was not harmed, since the mutant cannot grow on sialylated HMOs, this proves that these specific genes are required for its growth. It is unclear whether qPCR results of these additional genes will be helpful in our understanding of this process.

- b. The authors should also provide further rationale for deleting the 5 kb region encompassing the four genes. Why did they choose not to delete the entire cluster?

While deleting the entire cluster is more elegant, it is much more complicated, since it is almost 25kb long. We chose this specific region of these four genes, as it contains the transporter system and the unique sialidase (GH33) which was the top upregulated gene in the transcriptional analysis. We added a sentence explaining this point to the results section, L160-163: "We chose to delete this specific region as it contains the transport system and the unique GH33 sialidase which was the top upregulated gene in the transcriptional analysis".

- c. On a technical note, the complemented *B. dorei* strains grow to a lower final OD than the wild-type strain. Could the authors comment on potential explanations?

The complemented mutant reached a slightly lower OD than the WT strain, possibly due to the expression of the complemented genes from the plasmid's integration site, which differs from the endogenous genomic location of PUL32. We added a sentence commenting about this on L176-178: "The complemented mutant reached a lower OD than the WT strain, possibly due to the expression of the complemented genes from the plasmid's *att1/2* integration site, rather than their endogenous genomic location."

- d. The second line of evidence for sufficiency is that expression of the four genes from PUL32 allows *B. uniformis* and *B. stercoris* to grow in the presence of sialylated HMOs. That said, it is not clear how well the strains are growing. The authors should include growth traces in the presence of a preferred carbon source for each strain and/or lactose. In addition, the authors should note if these strains encode any homologs of the remaining genes in PUL32 or contain any PULs that are similar to PUL32.

We have added a growth curve of WT *B. uniformis* and *B. stercoris* in MM-lactose

to serve as a reference to how well these strains grow on preferred carbon sources (Fig 3, Supp. Fig 6). Since growth is not robust on lactose as well, the growth on sialylated HMOs that was conferred by p_compFull is comparable and significant, and we addressed this point in the text, L188-192: “The *B. uniformis* strain was able to fully grow on the sialylated glycans with the addition of the PUL32 core genes, to an OD600 comparable to that of lactose, and the *B. stercoris* isolate exhibited significant, yet modest growth on the sialylated glycans. Although the growth of *B. stercoris*::p_compFull on sialylated HMOs is not robust, it is noteworthy especially in the context of *B. stercoris*’ growth on lactose, which is moderate as well.” .

Also, we modified the text to include the information that *B. uniformis* and *B. stercoris* do not encode any PUL32 homologs, L184-185, and explained how homologs were searched for in the Methods section.

- e. The authors further claim that GH33 is the main gene that is required for use of sialylated HMOs. To further probe sufficiency, the authors should complement this gene on its own, in their *B. dorei* knockout strain as well as in the additional *Bacteroides* strains.

We thank the reviewer for this comment. We complemented the mutant *B. dorei* strain, as well as the *B. uniformis* and *B. stercoris* strains, with GH33 alone, and discussed the findings in the Results section, L211-218: “We further validated this result by complementing *B. dorei* Δ5k-PUL32 with GH33 only (p_compPartial4), which enabled growth on sialylated HMOs (**Fig 3B,C**). A similar pattern was observed for *B. uniformis* CL03T12C37, for which complementation with plasmids containing the GH33 gene (p_compPartial1/3/4) allowed modest growth on 3’-SL and 6’-SL, whereas complementation with the p_compPartial2 plasmid that lacks GH33 did not convey this ability (**Fig 3B,C**). However, for the *B. stercoris* isolate, none of the partial complementation plasmids were sufficient to drive significant growth on sialylated HMOs (**Supp. Fig 6**), suggesting additional functions are missing in the *B. stercoris* cellular environment to enable even a modest growth”.

- f. In their previous paper, the authors described that some *B. vulgatus* isolates are also able to grow on sialylated HMOs and that the *B. vulgatus* isolates also contain a GH33 that is upregulated in the presence of sialylated HMOs. Is this GH33 similar to the GH33 the authors focus on here and is it embedded in a PUL similar to PUL32 in these *B. dorei* isolates? This could provide further evidence that this GH33 is the important gene for breakdown of sialylated HMOs.

Thank you for your insightful comment, we addressed the GH33 protein sequence of *B. dorei* and *B. vulgatus*, and the conservation of PUL32 as its genomic context in the Discussion section, L272-279:

“The conservation of PUL33 across all *B. dorei* and *B. vulgatus* genomes deposited in the PULDB database suggests that PUL33 is important for the breakdown of sialylated HMOs. For *B. thetaiotaomicron* VPI 5482, a commonly used type strain, the closest PUL33 homolog is PUL9, experimentally shown to be involved in degrading sialic-acid containing mucins and HMO utilization. While the

GH33 protein sequence in the *B. dorei* and *B. thetaiotaomicron* clusters is similar (67.65%), their SusC transporters are only 20.67% (**Supp. Fig 9**), suggesting the transporters have evolved to recognize varying substrates, or that these two clusters evolved independently to perform similar functions in different ways.”

And L294-297:

“Finally, when we zoom into GH33, the key gene in PUL33 responsible for the cleavage of the sialic acid residue from the HMO backbone, we find that *B. dorei*'s GH33 protein sequence is very similar to that of its close relative *B. vulgatus* and less so to *B. thetaiotaomicron* (97.44% vs. 67.65% similarity; **Supp. Fig 9**). “

I realize this is a substantial amount of work, and the authors need not conduct all of it. For example, it may be acceptable to address points a and d. As a complement, it would be important to tone down claims of sufficiency for sialylated HMO use in the manuscript.

5. The authors state that GH33-PUL8 is insufficient to rescue growth on sialylated sugars and hypothesize that the gene has distinct functions from GH33-PUL32. It is possible that the gene has similar function but is not expressed under the growth conditions used in this study. The authors show that GH33-PUL8 is not significantly upregulated upon growth in sialylated HMOs but it is not clear what the absolute expression levels of the gene are. Is the GH33-PUL8 gene expressed at appreciable levels under the conditions used here?

Since our submission, an update to the CAZY database has affected the annotation structure of PUL8 specifically. It is now split into two distinct PULs (termed now PUL8 and PUL9), excluding the GKD17_02665 gene, previously annotated as GH33. This new version, which we have updated in our resubmitted manuscript, highlights the fact that the previously annotated GH33 indeed has other functions, as we predicted in our original manuscript. We have altered the text accordingly, and compared it as it was previously suspected to play a role, L139-144: “However, previous versions of *B. dorei*'s assembly included an additional GH33 sialidase named GKD17_02665 found in a previous version of PUL8. The expression of previously-annotated GH33 was not upregulated at all in our data (**Fig 1A,B**), supporting its non-GH33 annotation in current versions, and further emphasizing the importance and specificity of GH33 found in PUL32 in the utilization process of sialylated HMOs.”

In addition, using RT-qPCR, we also validated that in the WT strain that was later genetically modified, GH33-PUL8 (GKD17_02665) is not upregulated on sialylated HMOs, in contrast to GH33-PUL32. Please see Supp. Fig 4A.

6. The introduction makes the utilization of HMOs by *Bacteroides* seem unstudied, while also introducing research that has already shown other *Bacteroides* species as known HMO consumers. I would recommend toning down claims of complete lack of knowledge into this process.

We thank the reviewer for pointing this out. We toned-down the statements in the Abstract and Introduction, and elaborated more on the status of current knowledge and previous research regarding HMO utilization by *Bacteroides* in the introduction L62-67: “Some of the mechanisms have been further identified, for example, that *B. thetaiotaomicron* and

B. fragilis utilize both host mucin-O-glycans and HMOs using a similar set of upregulated genes. For fucosylated HMOs, fucosidases from *Bacteroides* were profiled and showed versatility in efficiency and substrate bond specificity. Finally, for sialylated sugars, one of *B. thetaiotaomicron*'s GH33 sialidases was crystallized and characterized, unraveling a wide binding groove accounting for a broad substrate specificity, including various sialylated HMOs."

Additional points:

- a. Line 53: "Taken together, all these suggest that *Bacteroides* species are the immediate next candidates for HMS utilization" hints at a hierarchy between genera that is not clear, and the phrasing undersells the previous work identifying *Bacteroides* as HMO consumers – it may be better to change the wording to emphasize that *Bacteroides* species are known HMO utilizers, but that many of the mechanisms remain unknown.

We altered the text to "all these suggest that *Bacteroides* species are also candidates for HMO utilization".

- b. Line 63-65: Please cite again references to which this statement refers.

We re-wrote these sections of the introduction to be more clear, and now specify and cite the HMO studies that were performed in *Bacteroides*, L62-67: "Some of the mechanisms have been further identified, for example, that *B. thetaiotaomicron* and *B. fragilis* utilize both host mucin-O-glycans and HMOs using a similar set of upregulated genes⁵⁶. For fucosylated HMOs, fucosidases from *Bacteroides* were profiled and showed versatility in efficiency and substrate bond specificity⁵⁸. Finally, for sialylated sugars, one of *B. thetaiotaomicron*'s GH33 sialidases was crystallized and characterized⁵⁹, unraveling a wide binding groove accounting for a broad substrate specificity, including various sialylated HMOs⁶⁰." Then we state the need for additional research on individual HMOs in that context, L68-70: "While these pioneering works lay the foundation for understanding HMO utilization by *Bacteroides* species, further research is needed to extend the mechanistic characterization to additional infant gut commensal strains, and specific types of HMOs."

Minor Comments

1. The authors validate their knockout strain using PCR with primers flanking the region targeted for knockout. Although this is a good approach in general, it is fraught for large deletions: it is impossible to detect if there are cells containing the wild-type genotype within the population, because the PCR will strongly favor the smaller product. I would recommend supplementing these data with PCRs with pairs of primers that uniquely recognize the junctions in the different genotypes. For example, one primer pair could be designed with one primer outside of the targeted region and one primer inside the targeted region, and demonstrating that this primer pair produces a product only in the wild-type strain.

We thank the reviewer for this suggestion, we performed an additional PCR reaction with one primer inside the deleted region and one primer outside of it, added the results to Fig

2C and addressed them in the text, L162-166: “To validate the mutant, we performed two PCR reactions: First, using primers located before and after the deletion, showing that the PCR product is 5,000bp shorter in the mutant strain compared to the wild type (WT) strain (P1, P2 in **Fig 2C**); Second, with one primer located within the deleted region, demonstrating a product at the predicted length for the WT and a lack of product for the mutant strain (P3, P4 in **Fig 2C**).“

2. The pink colors used to indicate 3-SL and 6-SL in the growth curves throughout the paper are very similar and difficult to differentiate. I suggest using two colors with a stronger contrast.

We adjusted the colors to make the distinction between 3'-SL and 6'-SL more clear.

3. Lines 80 and 144: I would recommend that the authors be careful with their use of “essential”, as many readers will interpret that as “essential for growth” rather than “essential for HMO utilization”. Alternative wording could include “important” or “necessary”.

We changed the use of “essential” to “necessary”

4. Line 67: Change “parts” to “tools”. Done.

5. Line 83: Delete “a”. Done.

6. Line 108: please note fold changes instead of log2 fold changes.

We thank the reviewer for this comment and agree that fold change values are easier to interpret. We changed the values to fold-change in the text.

7. Line 131: tone down claim of “fully restored”, as the complemented strains appear to have defects in growth compared to wild-type.

We toned down the “fully restored” statement to “restored” (L175) and addressed the complemented strain’s minor growth defect in the text, see response to major comment 4c.

8. Line 157: Please include the data in the supplemental figures.

Thank you, we included the data in Supp. Fig 7.

9. Especially in the discussion, please refer to the strains by their genotypes rather than as p_compPartial3 etc, e.g. “B. dorei strain lacking/expressing X”.

We improved phrasing in the discussion, referring to the strains by their genotypes, to make things clearer.

10. Figure 2A: please include the Cas12 gene in the schematic of the plasmid. Done.

11. Figure 2D: It is unclear how the statistical comparisons of the WT to $\Delta 5k$ strains, and $\Delta 5k$ to p_comp strains was performed. The methods states a paired t-test at two timepoints, but only provides a single p-value per carbon source.

Thank you for raising this point. The p-values reported in the figure are the values for the t-tests performed after 24 hours. We have added the full statistical data as a

supplementary table (Supp. Table 1), containing both the mean difference between the OD600 of the strains at the selected timepoints (1h + 24h), and the corresponding p-value. We have also explained this more clearly and clarified what p-values are reported in the Methods, L355-358:

“The p-values for the latter timepoints are reported in the main figures (**Fig 2D, 3A**), and the full statistical data at both timepoints, including the mean difference between the OD600 values of the growth curves, are reported in **Supplementary Table 1**. A mean difference > 0.1 with a p-value < 0.01 was considered statistically significant.”

12. Supp. Fig. 3: It is not clear what the difference between the two lactose/no carbon panels is. Additionally, it looks like mean lines are plotted in a slightly lighter shade along with the individual samples which makes the data hard to read.

In this figure (now Supp. Fig 5), the lactose/no carbon panels represent the positive and negative controls of the growth curve experiments, and the reason for the two panels is that the growth of the WT and mutant *B. dorei* strains was examined separately on fucosylated HMOs (top row) and neutral HMOs (middle row), each experiment having its own controls. Since the lactose controls indeed look almost identical, and following your comment, we modified the layout of the plot to make it clearer.

The shades were corrected for easier reading of the data.

REVIEWERS' COMMENTS

Reviewer #1 (Remarks to the Author):

While the authors have acknowledged the omissions and overstatements undermining the original manuscript, the revised version remains limited in scope and would require substantiate additional experimental work as suggested earlier.

Reviewer #2 (Remarks to the Author):

The authors have addressed most of my previous comments and the revised version looks excellent. I do have a few additional minor suggestions:

1. Use of “necessary and sufficient” in L193/194. The authors conclude sufficiency from the experiments transferring the 4-gene cluster into *B. uniformis* and *B. stercoris*. The strains are on their own unable to grow on sialylated HMOs but expression of the 4 genes allows them to grow. This is strong evidence. However the sentence as it is worded at present might be misunderstood. In particular it is very likely that other genes are also required for full breakdown of the sialylated HMOs, including the other genes in PUL33. It would be more accurate to state that the genes are sufficient to cleave the sialic acid group from sialylated HMOs and thereby enable growth on these HMOs.

2. The core of my previous comment about GH33-PUL8 (previous comment #5) was not addressed. In particular, the point was that lack of upregulation of GH33-PUL8 by the tested sialylated HMOs is insufficient to infer that the gene does not play a role in breakdown of sialylated HMOs in general. For example, it may not be expressed under the growth conditions tested here, such that it cannot play a role in the breakdown of the sialylated HMOs tested here. Further, it may well be upregulated by other sialylated HMOs and enable their breakdown. The new classification of GH33-PUL8 does make this less likely, but I would recommend the authors tone down their statement to not that they cannot rule out that GH33-PUL8 does not play a role in use of other sialylated HMOs not tested here, but that in the context of their HMOs GH33-PUL33 is the main sialidase. This is particularly important given that only a small number of HMOs were tested.

3. Data in Figure 3C and described in L212:

a. “a similar pattern” – the pattern does not really look similar between *B. dorei* and *B. uniformis*. It would be more appropriate to drop that phrase and simply describe the data.

b. In Figure 3C, the y-axis scale is inconsistent for the different panels, in two ways. First, the y-axis labels suggest that the *B. uniformis* data are plotted on a log scale and the *B. dorei* data plotted on a linear scale. This would be very unusual. Alternatively perhaps the axis labels are wrong. Among the *B. uniformis* panels, the y-axis scale looks different for p_compPartial4 compared to the other strains, as evidenced by the grid. Please ensure that the scales are similar and the axes appropriately labeled.

4. Finally, the manuscript could use additional proofreading. There were multiple instances of grammatical issues, erroneously placed or missing commas, and additional mistakes. A few examples that are not comprehensive:

a. L15/16 - Commas in inappropriate locations

b. L185 – “Homologes” should be changed to “Homologues” or “Homologs”

c. L290/293 – “sialic acid” and “sialic-acid” spelled differently within the same sentence.

d. Figure 3 Legend: “assays”, not “essays”

Response to reviewer

We would like to thank and appreciate the reviewer for their suggestions.

The authors have addressed most of my previous comments and the revised version looks excellent. I do have a few additional minor suggestions:

1. Use of “necessary and sufficient” in L193/194. The authors conclude sufficiency from the experiments transferring the 4-gene cluster into *B. uniformis* and *B. stercoris*. The strains are on their own unable to grow on sialylated HMOs but expression of the 4 genes allows them to grow. This is strong evidence. However the sentence as it is worded at present might be misunderstood. In particular it is very likely that other genes are also required for full breakdown of the sialylated HMOs, including the other genes in PUL33. It would be more accurate to state that the genes are sufficient to cleave the sialic acid group from sialylated HMOs and thereby enable growth on these HMOs.

We thank the reviewer for pointing this out. We changed the text as suggested to “These results indicate that the four core genes (GH20, GH33, SusD, and SusC) are sufficient to cleave the sialic acid group from sialylated HMOs and thereby enable growth on these sugars.”

2. The core of my previous comment about GH33-PUL8 (previous comment #5) was not addressed. In particular, the point was that lack of upregulation of GH33-PUL8 by the tested sialylated HMOs is insufficient to infer that the gene does not play a role in breakdown of sialylated HMOs in general. For example, it may not be expressed under the growth conditions tested here, such that it cannot play a role in the breakdown of the sialylated HMOs tested here. Further, it may well be upregulated by other sialylated HMOs and enable their breakdown. The new classification of GH33-PUL8 does make this less likely, but I would recommend the authors tone down their statement to not that they cannot rule out that GH33-PUL8 does not play a role in use of other sialylated HMOs not tested here, but that in the context of their HMOs GH33-PUL33 is the main sialidase. This is particularly important given that only a small number of HMOs were tested.

We thank the reviewer for further clarifying this point, and now state that the ability to make a conclusion about GH33-PUL8’s role in the breakdown of sialylated HMOs is limited by the small amount of HMOs tested.

We changed the phrasing from “The expression of previously-annotated GH33 was not upregulated at all in our data (Fig 1A,B), supporting its non-GH33 annotation in current versions, and further emphasizing the importance and specificity of GH33 found in PUL33 in the utilization process of sialylated HMOs.”

To “The expression of the previously-annotated GH33 was not upregulated at all in our data (Fig 1A,B). The lack of upregulation of GH33-PUL8 under the tested growth conditions supports the notion that GH33-PUL33 in the main sialidase in the context of 3’-SL and 6’-SL, however does not rule out its potential role in utilizing other sialylated HMOs not tested here.”

3. Data in Figure 3C and described in L212:

a. “a similar pattern” – the pattern does not really look similar between *B. dorei* and *B. uniformis*. It would be more appropriate to drop that phrase and simply describe the data.

b. In Figure 3C, the y-axis scale is inconsistent for the different panels, in two ways. First, the y-axis labels suggest that the *B. uniformis* data are plotted on a log scale and the *B. dorei* data plotted on a linear scale. This would be very unusual. Alternatively perhaps the axis labels are wrong. Among the *B. uniformis* panels, the y-axis scale looks different for p_compPartial4 compared to the other strains, as evidenced by the grid. Please ensure that the scales are similar and the axes appropriately labeled.

Thank you for pointing this out. We discarded the phrase “a similar pattern” and corrected the axis in Figure 3C in which the ticks were erroneously placed (it is not logarithmic).

4. Finally, the manuscript could use additional proofreading. There were multiple instances of grammatical issues, erroneously placed or missing commas, and additional mistakes. A few examples that are not comprehensive:

- a. L15/16 - Commas in inappropriate locations
- b. L185 – “Homologes” should be changed to “Homologues” or “Homologs”
- c. L290/293 – “sialic acid” and “sialic-acid” spelled differently within the same sentence.
- d. Figure 3 Legend: “assays”, not “essays”

We thank the reviewer and revised the manuscript text where appropriate.